# Sequential Causal Discovery with Noisy Language Model Priors

**Prakhar Verma**\*                                                                 *prakhar.verma@aalto.fi*
*ELLIS Institute Finland and Aalto University*

**David Arbour**                                                                       *arbour@adobe.com*
*Adobe Research*

**Sunav Choudhary**                                                               *schoudha@adobe.com*
*Adobe Research*

**Harshita Chopra**†                                                                  *hchopra3@uw.edu*
*University of Washington, Seattle*

**Arno Solin**                                                                         *arno.solin@aalto.fi*
*ELLIS Institute Finland and Aalto University*

**Atanu R. Sinha**                                                                      *atr@adobe.com*
*Adobe Research*

**Reviewed on OpenReview:** *https://openreview.net/forum?id=wFs71JzEO7*

## Abstract

Causal discovery from observational data typically assumes access to complete data and availability of perfect domain experts. In practice, data often arrive in batches, are subject to sampling bias, and expert knowledge is scarce. Language Models (LMs) offer a surrogate for expert knowledge but suffer from hallucinations, inconsistencies, and bias. We present a hybrid framework that bridges these gaps by adaptively integrating sequential batch data with LM-derived noisy, expert knowledge while accounting for both *data-induced* and *LM-induced* biases. We propose a representation shift from Directed Acyclic Graph (DAG) to Partial Ancestral Graph (PAG), that accommodates ambiguities within a coherent framework, allowing grounding the *global* LM knowledge in *local* observational data. To guide LM interactions, we use a sequential optimization scheme that adaptively queries the most informative edges. Across varied datasets and LMs, we outperform prior work in structural accuracy and extend to parameter estimation, showing robustness to LM noise.

## 1 Introduction

Inference of causal relations from observational data remains a challenge in applications across healthcare, economics, business, and scientific discovery (Sanchez et al., 2022; Tu et al., 2019; Sadeghi et al., 2023; Ebert-Uphoff & Deng, 2012). The challenge is addressed through a dual approach: applying causal learning algorithms to observational data while incorporating domain expertise to resolve structural uncertainties (Spirtes et al., 2000; Neapolitan et al., 2004; Spirtes & Zhang, 2016; Chickering, 2002). However, domain expertise can be a scarce resource (He & Geng, 2008; Choo & Shiragur, 2023; Mooij et al., 2016; Meek, 2013; Constantinou et al., 2023). Advanced Language Models (LMs) create opportunities to explore their potential as surrogate experts for causal discovery (Kiciman et al., 2024; Willig et al., 2022). LMs generate informative

---

\*Work done during an internship with Adobe Research
†Work done while the author was with Adobe Research

Table 1: **LMs are overly optimistic:** LM based DAG-Pairwise and DAG-triplet prompting methods achieve high recall with low precision across temperatures on two common causal discovery datasets. This limitation calls for explicitly modeling data and LM biases. SHD=Structural Hamming Distance.

| Dataset | Temp. | Method | GPT-3.5$_{turbo}$ | | | GPT-4o | | |
|---|---|---|---|---|---|---|---|---|
| | | | SHD ↓ | Precision ↑ | Recall ↑ | SHD ↓ | Precision ↑ | Recall ↑ |
| EARTHQUAKE | 0.0 | Pairwise | $3.0_{\pm 0.0}$ | $0.57_{\pm 0.00}$ | $1.0_{\pm 0.0}$ | $2.0_{\pm 0.0}$ | $0.67_{\pm 0.00}$ | $1.0_{\pm 0.0}$ |
| | | Triplet | $2.1_{\pm 0.3}$ | $0.66_{\pm 0.03}$ | $1.0_{\pm 0.0}$ | $4.8_{\pm 0.4}$ | $0.46_{\pm 0.02}$ | $1.0_{\pm 0.0}$ |
| | 0.5 | Pairwise | $2.8_{\pm 0.6}$ | $0.59_{\pm 0.05}$ | $1.0_{\pm 0.0}$ | $2.2_{\pm 0.4}$ | $0.65_{\pm 0.04}$ | $1.0_{\pm 0.0}$ |
| | | Triplet | $2.1_{\pm 0.3}$ | $0.66_{\pm 0.03}$ | $1.0_{\pm 0.0}$ | $4.4_{\pm 0.5}$ | $0.48_{\pm 0.03}$ | $1.0_{\pm 0.0}$ |
| | 1.0 | Pairwise | $3.2_{\pm 0.6}$ | $0.56_{\pm 0.05}$ | $1.0_{\pm 0.0}$ | $1.8_{\pm 0.4}$ | $0.69_{\pm 0.05}$ | $1.0_{\pm 0.0}$ |
| | | Triplet | $2.0_{\pm 0.6}$ | $0.67_{\pm 0.07}$ | $1.0_{\pm 0.0}$ | $6.2_{\pm 0.8}$ | $0.39_{\pm 0.03}$ | $1.0_{\pm 0.0}$ |
| ASIA | 0.0 | Pairwise | $25.2_{\pm 0.4}$ | $0.24_{\pm 0.00}$ | $1.0_{\pm 0.0}$ | $11.2_{\pm 0.4}$ | $0.42_{\pm 0.01}$ | $1.0_{\pm 0.0}$ |
| | | Triplet | $24.5_{\pm 0.5}$ | $0.25_{\pm 0.00}$ | $1.0_{\pm 0.0}$ | $19.0_{\pm 0.6}$ | $0.30_{\pm 0.01}$ | $1.0_{\pm 0.0}$ |
| | 0.5 | Pairwise | $23.4_{\pm 1.0}$ | $0.26_{\pm 0.01}$ | $1.0_{\pm 0.0}$ | $11.6_{\pm 0.5}$ | $0.48_{\pm 0.01}$ | $1.0_{\pm 0.0}$ |
| | | Triplet | $24.0_{\pm 1.0}$ | $0.25_{\pm 0.01}$ | $1.0_{\pm 0.0}$ | $19.2_{\pm 0.8}$ | $0.29_{\pm 0.01}$ | $1.0_{\pm 0.0}$ |
| | 1.0 | Pairwise | $23.2_{\pm 1.5}$ | $0.25_{\pm 0.02}$ | $0.9_{\pm 0.1}$ | $11.8_{\pm 0.4}$ | $0.40_{\pm 0.01}$ | $1.0_{\pm 0.0}$ |
| | | Triplet | $23.3_{\pm 1.0}$ | $0.26_{\pm 0.01}$ | $1.0_{\pm 0.0}$ | $26.2_{\pm 1.7}$ | $0.23_{\pm 0.01}$ | $1.0_{\pm 0.0}$ |

Overly optimistic behavior of the LM experts lead to high recall and low precision.

✗ No *global* causal structure
✗ No grounding to *local* data
✗ Need heuristics to postprocess

priors or contraints (Takayama et al., 2025; Long et al., 2022; Ban et al., 2023b), improving accuracy when combined with data-driven algorithms. Yet, LMs pose their own challenges: hallucination, inconsistency, or failure to capture context-specific nuances (Ji et al., 2023; Kiciman et al., 2024).

The challenges compound since in common applications, observational data arrive batch-wise at a cadence, instead of as a complete dataset. Examples include web and app metrics of all online firms, where, for example, data could arrive weekly. Privacy regulations and storage constraints may further restrict data access to a short look-back window. A given week's (batch) data may not be representative of the overall distribution, which we assume remains stationary. This kind of arrival introduces *data-induced* bias, since the non-random draw of a batch suffers from sample selection bias (hereafter, selection bias Spirtes et al. (1995)) that distorts causal discovery. Separately, use of LMs, including *large* ones, poses two problems: *(i)* As surrogates for domain expertise, LMs introduce an *LM-induced* bias—their responses in terms of informative causal priors are prone to hallucinations, contextual brittleness, and inconsistency (Ji et al., 2023; Kiciman et al., 2024). *(ii)* The *global* knowledge encoded in LMs may not align with domain-specific *local* patterns emblematic of batch-wise data, leading to potentially biased learning.

Inattention to the dual biases—data-induced and LM-induced—is a key gap in current approaches to causal discovery with LMs, which we address. First, we propose a change in representation shift from a Directed Acyclic Graph (DAG), which LM-augmented causal discovery methods currently use, to a Partial Ancestral Graph (PAG), to accommodate uncertainty in the causal structure arising from the dual biases. Second, we propose a novel Bayesian-inspired approach to causal structure discovery, where beliefs over causal structure are updated with new data-batch, while augmenting noisy LM-knowledge as priors In support of PAG, we show that popular methods of pairwise and triplet prompting are *overly-optimistic* (*cf.* Table 1) and generate unreliable causal structure in the form of a DAG.

We introduce NLPSCM (Noisy Langue Prior in Sequential Causal Modeling); see Fig. 1 for an overview of the framework. NLPSCM differs from existing methods that either rely solely on access to the complete observational data or treat LMs as primary discovery mechanism. In a novelty, the causal structure discovery itself is Bayesian-inspired in that the beliefs about causal structure from data are updated iteratively by information drawn from an LM, as data arrive in batches. That is, we adopt a *data-first* approach, where for each batch, a traditional causal discovery algorithm, such as FCI from Spirtes et al. (1995), constructs an initial PAG conditioned on the background knowledge, which is then iteratively refined through optimized LM queries that leverage the global knowledge while remaining grounded in observed data. To maximize performance under limited budget, LM interactions are framed as a *sequential optimization* problem, selecting the most important edges to query, while accumulating background knowledge over batches. Moreover, to complete the causal discovery process, the parameter (edge weights) estimation we propose is *also* Bayesian, which incorporates potentially noisy LM priors on latent confounders and causal relationships. Our method uncovers cross-sectional causal relationships within each batch, rather than modeling temporal dependencies. NLPSCM applies especially to situations where studying contemporaneous causal relations among metrics is separated from confounding temporal effects.

**Contributions**   We summarize the contributions as follows: *(i)* We propose a representation shift from DAGs to PAGs, in a hybrid setup of batch, observational data and LM as noisy expert, that inherently captures uncertainty in causal structure learning. *(ii)* A Bayesian-inspired algorithm for *causal structure* discovery with sequential batch data treating LMs as noisy experts thus accounting for dual sources of bias. *(iii)* A *sequential optimization* strategy for selecting maximally informative LM edge queries under fixed LM budget constraints. *(iv)* A Bayesian *parameter estimation* algorithm that robustly integrates noisy LM priors with batched data. Taken together, these constitute  NLPSCM—an end-to-end causal discovery framework that jointly addresses both *causal structure learning* and *parameter estimation.*

## 2   Literature Review

**Traditional causal discovery**   Traditional causal discovery aims to recover the underlying causal structure from observational data by exploiting statistical dependencies, often formalized through graphical models such as DAGs and PAGs (Spirtes & Zhang, 2016; Pearl, 2009; Neapolitan et al., 2004). These approaches include constraint-based methods (*e.g.,* PC from Spirtes et al. (2000), FCI (Spirtes et al., 1995; Zhang, 2008), RFCI from Colombo et al. (2012)), score-based methods (*e.g.,* GES from Chickering (2002)), and functional causal models (*e.g.,* LiNGAM (Shimizu et al., 2006; 2011)), typically assuming causal sufficiency and faithfulness (Spirtes et al., 2000; Zhang & Spirtes, 2012). They rely on conditional independence tests or likelihood-based scoring to causal relationships (Shah & Peters, 2020; Zhang et al., 2011; Peters et al., 2017; Glymour et al., 2019). However, these approaches often struggle under data scarcity, presence of latent confounders (Spirtes et al., 2000; Monti et al., 2020), or domain-specific constraints not captured by statistical patterns (Mooij et al., 2016; Peters et al., 2014). These limitations are addressed by hybrid methods incorporating external domain knowledge (Meek, 1995; Heckerman et al., 1995; Ogarrio et al., 2016) from human experts, refining causal graphs with new variables, modifying edge orientations, or resolving equivalence classes (Brouillard et al., 2020; Wang et al., 2017; Constantinou et al., 2023; Ban et al., 2023b). This helps restrict the search space and improves identifiability, particularly when data is limited, as shown by Wallace et al. (1996). The evolution from purely statistical methods to knowledge-augmented approaches fuels advanced machine learning techniques to enhance causal discovery (Glymour et al., 2019; Schölkopf et al., 2021). Also, as noted by Baldi & Shahbaba (2020), causal research distinguishes general *vs.* local settings and applies to diverse fields (Andrade & Zachariadis, 2016; Bilal & Känzig, 2024; Geist & Lambin, 2002; Kelly et al., 2011; McKinney et al., 2016; Mathers et al., 2009).

Causal discovery extends to streaming data of networks, stock markets, and sensor systems. Causal Bayesian learning and causal discovery with progressively streaming features are well studied (Darvish Rouhani et al., 2018; Yu et al., 2010; You et al., 2023; Li et al., 2021; You et al., 2021; Yu et al., 2010). Instead, we focus on *contemporaneous* relationships among a fixed set of variables, which yield data across batches. In batch-wise experimentation in medicine and A/B testing, Bridgeford et al. (2021) examines associations between batches, with batch effects as causal effects, while Zhang & Yuan (2024) examines adaptive batch-wise intervention. We confine to sequential, batch-wise observational data.

**LM-augmented causal structure discovery**   LM augmented causal discovery relies on LM's world knowledge and includes pairwise prompting (Willig et al., 2022; Long et al., 2022; Kiciman et al., 2024; Jin et al., 2024; Long et al., 2023), and triplet-based prompting incorporating voting proposed by Vashishtha et al. (2025). Hybrid frameworks integrate LM-generated insights in constraint-based methods or inform score-based approaches (Ban et al., 2023b; Takayama et al., 2025). Reliability of LM-derived constraints is sensitive to domain specificity and prompt framing (Kiciman et al., 2024; Ji et al., 2023). Ban et al. (2023a) proposes ILS-CSL involving iterative refinement of causal graphs by alternating between LM reasoning and statistical verification. Augmenting LM prompts with correlation matrices or statistical summaries is explored (Jiralerspong et al., 2024; Susanti & Färber, 2025). Yet, LM-aided causal discovery shows inconsistent judgments across prompting strategies, difficulties with grounding, and biases. Recently, da Silva et al. (2025) proposed the BFS method. It is designed for full dataset while our method is anchored in sequential batched data. Additionally, its update is around information gain and does not perform edge-weight estimation.

Deviating from the prior art, we present a Bayesian-inspired framework for causal structure discovery, where data arrive sequentially in batches, and we recognize dual uncertainties—arising from limited observational data and noisy LM responses. We depart from DAG-centric discovery to adopt the PAG, a more robust

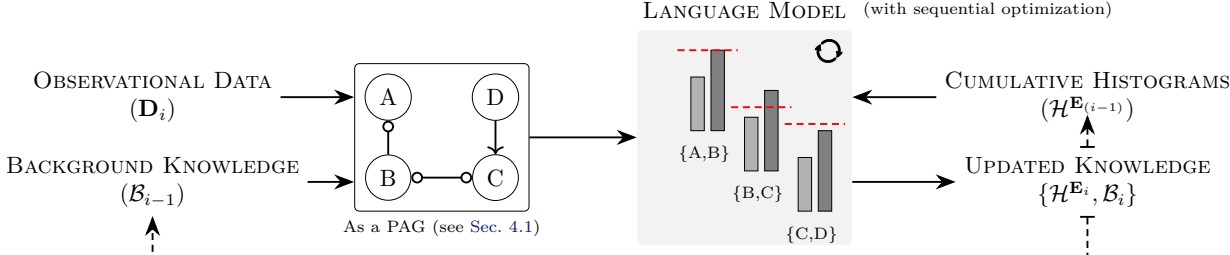

Figure 1: **Overview of the NLPSCM framework:** At each batch $\mathbf{D}_i$, observational data is combined with accumulated background knowledge $\mathcal{B}_{(i-1)}$ as prior to estimate a PAG structure. A language model is then queried—under sequential optimization—to produce beliefs over possible causal relations and update $\mathcal{B}_i$. The updated $\{\mathcal{H}^{\mathbf{E}_i}, \mathcal{B}_i\}$ are fed back into the next iteration.

representation for evolving, uncertain causal structures. We iteratively refine the PAG across batches, while framing LM queries as a sequential optimization problem.

**Parameter estimation in SEMs** While existing hybrid methods that combine observational data with language models (LMs) primarily focus on causal structure discovery, they typically stop short of estimating causal effect parameters, which we refer to as *parameter estimation*. In contrast, prior work on parameter estimation is predominantly framed within structural equation models (SEMs) and generally assumes either a known structure or purely data-driven priors.

Within the SEM framework, under assumptions of linearity and Gaussian noise (Bollen, 1989; Shimizu et al., 2006), classical approaches estimate parameters using maximum likelihood or two-stage least squares method, as described in Pearl (2009). Learning of SEM parameters extends to nonlinear or nonparametric settings using neural networks or Gaussian processes, enabling flexible modeling of complex dependencies while retaining causal interpretability (Zheng et al., 2020; Lachapelle et al., 2020), and helps in integrating expert knowledge with data-driven estimation(Peters et al., 2017; Schölkopf et al., 2021). However, the use of LM-derived priors and correlations for SEM parameter estimation remains largely unexplored.

To bridge this gap and provide an end-to-end causal discovery framework, we jointly perform causal structure discovery and SEM parameter estimation. The latter constitutes a key contribution of our work: we propose a principled parameter estimation procedure that integrates LM-derived noisy priors into SEMs, yielding consistent estimators of causal strengths despite misspecification of priors.

## 3 Problem Setup: Sequential Causal Discovery with LMs

Traditional causal discovery methods uncover causal structure by exploiting statistical dependencies in observational data, typically assuming access to the complete dataset, and reliable domain knowledge. In contrast, we focus on the setting of sequential, batch-wise observational data. This setting introduces dual sources of bias: *(i)* potentially biased and limited batched observational data, and *(ii)* noisy LM responses. We assume the underlying population distribution $p(X)$ is stationary across batches. However, we allow for selection bias, where each batch may not be drawn randomly from the population distribution. That is, for a batch $i$, $p(X \mid batch = i) \neq p(X)$. Below we introduce the notation and the problem setup.

**Problem statement** Given sequential batches of observational data subject to selection bias, we consider the problem of: *(i)* causal discovery in the presence of latent confounders, *(ii)* incorporating noisy priors provided by LMs, and *(iii)* parameter estimation of the inferred causal structure. We now define the problem formally.

**Notation and setup** We define batches of observational data $\mathcal{D} = \{\mathbf{D}_i\}_{i=0}^N$, where each $i^{\text{th}}$ batch is a sample from the same underlying *'true'* distribution, $\mathbf{D}_i \sim \mathbb{D}$. Each batch contains same set of observed variables, $\mathbf{V}^{\text{O}}$, with $\mathbf{D}_i \in \mathbb{R}^{n_i \times d}$, where $n_i$ is the number of data points, varies by batch, and $d = |\mathbf{V}^{\text{O}}|$ is the number of observed variables. $\mathbb{R}$ denotes real numbers. Any categorical value for an observed variable is encoded as a numerical value. All notations are succinctly shown in Table A1.

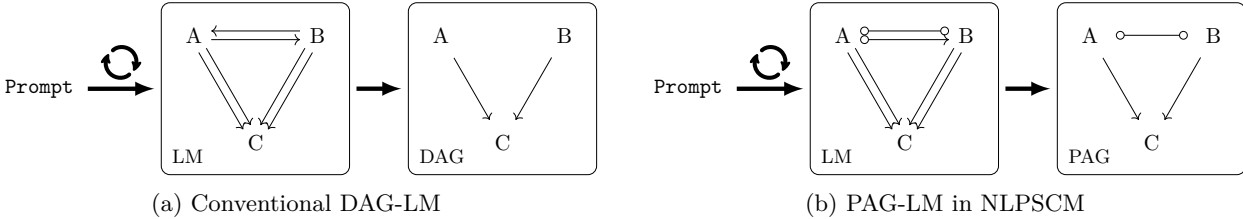

(a) Conventional DAG-LM            (b) PAG-LM in NLPSCM

Figure 2: **PAG-LM** streamlines how LMs compose the graph and allows for ambiguities to be indicated in the structure. *(a)* DAG is constructed by iterative prompting (⟳) leading to ambiguities (*e.g.,* ⇆) requiring heuristics that cannot be represented. *(b)* NLPSCM represents the causal structure as a PAG that implicitly allow ambiguities to be represented providing a richer representation ( *e.g.,* ∘–∘)

For each batch $\mathbf{D}_i$, a causal graph $\mathcal{G}^{D_i}=(\mathbf{V}^O, \mathbf{E}^{D_i})$ is inferred using a standard causal discovery algorithm, where $\mathcal{G}^{D_i}$ is a PAG with $\mathbf{V}^O$ nodes and $\mathbf{E}^{D_i}$ edges. We assume there exists a true causal graph $\mathcal{G}=(\mathbf{V}, \mathbf{E})$, where $\mathbf{V}^O \subseteq \mathbf{V}$ and $\mathbf{V}=\{\mathbf{V}^O, \mathbf{V}^L\}$ represents all the variables, both observed and unobserved latent, and $\mathbf{E}$ represents the true causal relationships. We assume that each confounder may affect two observed variables.

**LM-augmented sequential causal discovery** Traditional hybrid approaches rely on domain experts to narrow the gap between the inferred causal graph $\mathcal{G}^{D_i}$ and the true causal graph $\mathcal{G}$ by either introducing knowledge about unobserved variables, effectively reducing the set $\mathbf{V} \setminus \mathbf{V}^O$, and adding, removing, reorienting edges in the inferred graph, aiming to minimize the difference $\mathbf{E} \setminus \mathbf{E}^{D_i}$.

NLPSCM's hybrid approach extends the prior art in two directions: *(i)* using an LM as a noisy expert to improve the causal structure $\mathcal{G}^{D_i}$, obtained via known causal discovery algorithm, where *(ii)* data arrive sequentially in batches. The LM (noisy e**X**pert), represented $X^i$, helps in reducing the Markov equivalence class to yield causal structure ($\mathcal{G}^{X_i}$). Formally,

$$f_{CD} : \mathbf{D}_i \to \mathcal{G}^{D_i} \; ; \; f_{LM} : \mathcal{G}^{D_i} \to \mathcal{G}^{X_i} \,, \tag{1}$$

where $\mathcal{G}^{D_i} = (\mathbf{V}^O, \mathbf{E}^{D_i})$ and $\mathcal{G}^{X_i} = (\mathbf{V}^{X_i}, \mathbf{E}^{X_i})$. The aim is to reduce the discrepancy between the inferred graph and the true causal structure through LM expertise,

$$\mathcal{G} \setminus \mathcal{G}^{X_i} \leq \mathcal{G} \setminus \mathcal{G}^{D_i} \; ; \; \mathbf{V} \setminus \mathbf{V}^{X_i} \leq \mathbf{V} \setminus \mathbf{V}^O \; ; \; \mathbf{E} \setminus \mathbf{E}^{X_i} \leq \mathbf{E} \setminus \mathbf{E}^{D_i} \,. \tag{2}$$

The $\mathbf{V}^{X_i} \setminus \mathbf{V}^O$ is the set of *LM-suggested* variables while $\mathbf{E}^{X_i} \setminus \mathbf{E}^{D_i}$ is the set of *LM-suggested* edges. To integrate the LM's noisy responses and address the inherent bias in batch data, our Bayesian-inspired causal discovery framework explicitly handles both *data-induced* and *LM-induced* biases.

**Parameter estimation** Once the LM-augmented causal structure $\mathcal{G}^{X_i}$ is obtained, we focus on estimating the parameters of the structure equation model (SEM) *i.e.* the edge weights $\boldsymbol{\theta}$ and the noise variance $\sigma^2$; $\boldsymbol{\phi} = \{\boldsymbol{\theta}, \sigma^2\}$. A straightforward method to learn the parameters $\boldsymbol{\phi}$ is Maximum Likelihood Estimation (MLE), $\nabla_{\boldsymbol{\phi}} \log p(\mathbf{D}_i \,|\, \mathcal{G}^{X_i}, \boldsymbol{\phi})$. However, a critical limitation appears in the presence of unobserved *expert-suggested* variables ($\mathbf{V}^L$) in the augmented causal structure, as the likelihood becomes intractable. We address this by proposing a Bayesian parameter estimation algorithm that incorporates the *expert-suggested* information about the unobserved (latent) variable(s), $\mathbf{V}^L$.

## 4 Sequential Causal Structure Discovery with PAGs

The promise of LMs as proxies for domain expert in causal structure discovery, as studied in prior art, typically queried via pairwise or triplet prompts, faces key limitations: *(i)* LMs may provide responses regardless of causal relations among other variables, resulting in inconsistent or cyclic causal graphs; *(ii)* LMs may hallucinate; *(iii)* LMs may be overly optimistic, predicting spurious causal relationships with high recall but low precision (*cf.* Table 1). While prior art addresses *(i)* and *(ii)* through heuristics or auxiliary models to enforce acyclicity and consistency, they add complexity and potentially degrading performance. Crucially, *(ii)* and *(iii)* remain largely unexplored in the setting of sequential batch data where two distinct sources of bias emerge: *LM-induced* bias and *data-induced* bias.

## 4.1 PAG to Incorporate Uncertainty

Prompting methods (pairwise or triplet) constrain the response format to—*'causal', 'non-causal',* or *'unknown'*—which prevent LMs from expressing uncertainty, thus exacerbating bias and inconsistency. To address this, we propose a representational shift from DAG to PAG when using LMs as proxy for experts. PAGs encode uncertainty and structural ambiguity in a principled manner, accommodating latent confounding and partial orientation (*cf.* Fig. 2). Formally, we expand the limited edge set of DAG, $\mathbf{E}^{\text{DAG}} = \{\rightarrow, \leftarrow, \cdot\}$, to the richer set for PAG, $\mathbf{E}^{\text{PAG}} = \{\rightarrow, \leftarrow, \leftrightarrow, \circ\!\!\rightarrow, \leftarrow\!\!\circ, \circ\!\!-\!\!\circ, \cdot, -\}$ [1], where $\cdot$ represents no causal relation. The expansion allows LM to select from more options and improve causal discovery (*cf.* Table 2). In PAG-pairwise, the LM is queried per variable pair to select from $\mathbf{E}^{\text{PAG}}$ (see Sec. F.2 for the full prompt).

Table 2: **DAG to PAG:** Structural Intervention Distance (SID) between DAG-Pairwise (Voting) and PAG-Pairwise depicts benefit of PAG to represent inherent causal uncertainty.

| Dataset | Temp. | Method | SID ↓ |
|---|---|---|---|
| EARTHQUAKE | 0.0 | Pairwise (Voting) | $(1.0, 1.0)_{\pm (0.0,\, 0.0)}$ |
| | | PAG-Pairwise | $(0.0, 0.0)_{\pm (0.0,\, 0.0)}$ |
| | 0.5 | Pairwise (Voting) | $(1.4, 1.4)_{\pm (1.2,\, 1.2)}$ |
| | | PAG-Pairwise | $(0.0, 0.0)_{\pm (0.0,\, 0.0)}$ |
| | 1.0 | Pairwise (Voting) | $(1.6, 1.6)_{\pm (1.5,\, 1.5)}$ |
| | | PAG-Pairwise | $(0.8, 0.8)_{\pm (1.6,\, 1.6)}$ |
| ASIA | 0.0 | Pairwise (Voting) | $(6.4, 6.4)_{\pm (0.8,\, 0.8)}$ |
| | | PAG-Pairwise | $(3.8, 3.8)_{\pm (0.9,\, 0.9)}$ |
| | 0.5 | Pairwise (Voting) | $(5.2, 5.2)_{\pm (1.7,\, 1.7)}$ |
| | | PAG-Pairwise | $(3.5, 3.5)_{\pm (0.8,\, 0.8)}$ |
| | 1.0 | Pairwise (Voting) | $(5.4, 5.4)_{\pm (1.8,\, 1.8)}$ |
| | | PAG-Pairwise | $(4.0, 4.0)_{\pm (1.9,\, 1.9)}$ |

The representational shift to PAG is a necessary pivot and starting point for addressing dual sources of bias—*LM-induced* and *data-induced*. The LM's error prone response constitutes a prior and motivates a novel Bayesian causal structure discovery framework in the sequential batch setting, whereby we integrate causal predictions from observational data and LMs in a sequential, iterative manner. Building on this key observation, we introduce the causal structure learning algorithm of NLPSCM and then on the inferred structure introduce the parameter estimation algorithm.

## 4.2 LM-augmented Causal Structure Discovery

As a principled way to incorporate the dual sources of bias, we adopt a Bayesian-inspired formulation. Formally, given data batch $\mathbf{D}_i$, cumulative background knowledge $\mathcal{B}_{(i-1)}$ up to batch $(i-1)$, the posterior over $\mathcal{G}_i$ is obtained via Bayes' rules,

$$\underbrace{p(\mathcal{G}_i \,|\, \mathbf{D}_i, \mathcal{B}_{(i-1)})}_{\text{Posterior}} \propto \underbrace{p(\mathbf{D}_i \,|\, \mathcal{G}_i)}_{\text{Likelihood}} \underbrace{p(\mathcal{G}_i \,|\, \mathcal{B}_{(i-1)})}_{\text{Prior}} . \tag{3}$$

The prior $p(\mathcal{G}_i \,|\, \mathcal{B}_{(i-1)})$ is iteratively shaped by background knowledge, which accumulates across batches. Intuitively, the prior encodes the edge type between node pairs in the causal structure. Once the posterior is obtained, an LM is queried to update and obtain background knowledge $\mathcal{B}_i$ using Eq. (5) and Eq. (9), which shapes the prior for the next batch. We note that this formulation serves as a conceptual motivation for the design of NLPSCM; the implementation maintains uncertainty at the edge level (Eqs. (4) and (5)) rather than computing a full posterior over graph space. Next we discuss the formulation.

Given the inherent stochasticity of LMs, its response can be viewed as a *sample* from an implicit distribution over the edge types ($\mathbf{E}^{\text{PAG}}$). This allows explicit modeling of uncertainty in LM responses, $f_{\text{LM}}^{(i)}(A, B, \text{Pr}) \sim p(E_{AB} \,|\, A, B)$, where $A, B$ are the two nodes, Pr is the prompt, and $E_{AB} \in \mathbf{E}^{\text{PAG}}$. Treating LM responses as noisy observations rather than ground truth, addresses the challenge of LM hallucination. As more batches are processed, the prompt Pr becomes more informative, thereby decreasing uncertainty in LM responses. Formally, we treat LM as a *black-box causal edge sampler* and aggregate multiple LM samples into empirical histograms that are updated iteratively over batches,

$$\mathcal{H}^{\mathbf{E}_i}(A, B)[E_{AB}] = \mathcal{H}^{\mathbf{E}_{i-1}}(A, B)[E_{AB}] + \mathbb{I}[f_{\text{LM}}^{(i)}(A, B) = E_{AB}], \tag{4}$$

---

[1]To capture selection bias, ambiguous edges ($\circ\!\!-\!\!\circ, \circ\!\!\rightarrow$) and undirected edges '$-$' can be used. Not all causal discovery algorithms output undirected edges (*e.g.,* https://causal-learn.readthedocs.io/en/latest/search_methods_index/Constraint-basedcausaldiscoverymethods/FCI.html), but they output ambiguous edges, which we use as proxy for selection bias.

Table 3: **Semantic Entropy in NLPSCM.** We report the mean Shannon entropy of empirical histograms over PAG edge predictions (mean ± std over 5 runs) across sequential batches for USER LEVEL DATA–I and USER LEVEL DATA–II. The observed reduction in entropy indicates decreasing predictive uncertainty over PAG edge types as structural context is progressively incorporated.

| Dataset | Batch-1 | Batch-2 | Batch-3 | Batch-4 | Batch-5 | Batch-6 | Batch-7 |
|---|---|---|---|---|---|---|---|
| USER LEVEL DATA - I | $0.48 \pm 0.06$ | $0.50 \pm 0.05$ | $0.36 \pm 0.08$ | $0.17 \pm 0.13$ | $0.10 \pm 0.06$ | $0.12 \pm 0.06$ | $0.09 \pm 0.05$ |
| USER LEVEL DATA - II | $0.31 \pm 0.25$ | $0.27 \pm 0.22$ | $0.18 \pm 0.18$ | $0.12 \pm 0.19$ | $0.07 \pm 0.15$ | $0.00 \pm 0.00$ | $0.00 \pm 0.00$ |

where $\mathcal{H}^{\mathbf{E}_i}(A, B)$ represents the cumulative histogram up to batch $i$, and $E_{AB}$ represents a type of causal relation. These histograms define an approximate posterior distribution over edge types, capturing the LM's evolving beliefs about causal relationships.

To determine when accumulated LM evidence is sufficient to promote an edge to background knowledge, we define a dynamic threshold that balances distributional uncertainty and sampling uncertainty (motivated by the explore-exploit trade-off formalized in Sec. 4.3),

$$\tau_i^e = \alpha \times E_i^e \times T_i^e + (1 - \alpha)\sqrt{T_i^e\left(1 - \frac{T_i^e}{T_i}\right)}, \quad \text{s.t.} \quad E_i^e = -\sum_j \frac{\mathcal{H}_{j,e}^{\mathbf{E}_i}}{T_i^e} \times \log\left(\frac{\mathcal{H}_{j,e}^{\mathbf{E}_i}}{T_i^e}\right). \tag{5}$$

Here, for batch $i$ and edge $e$, $\tau_i^e$ denotes the threshold, $E_i^e$ the posterior entropy, $T_i^e$ the number of LM interactions, $T_i = \sum_e T_i^e$ the total interactions, and $\mathcal{H}_{j,e}^{\mathbf{E}_i}$ the frequency of bin $j$ in $e$'s histogram.

Intuitively, the first term in $\tau_i^e$ of Eq. (5) accounts for uncertainty in the histogram edge distribution, while the second term handles the sampling uncertainty that decreases as more batches of data arrive. The hyperparameter $\alpha$ balances between these two terms. Fig. 4 showcases the efficiency of the proposed dynamic background threshold (*cf.* Eq. (5)), with the additional details discussed in Sec. 6.1. The pseudo-code of the algorithm is outlined in Alg. 1.

A key property of our sequential framework is that the prompt Pr becomes progressively more informative across batches as $\mathcal{G}^{X_i}$ expands, providing increasingly rich structural context to the LM. This corresponds to conditioning on accumulated causal constraints, and therefore the model's predictive uncertainty is expected to decrease over iterations. We empirically validate this behavior by reporting the mean histogram entropy across batches for the two real world datasets USER LEVEL DATA–I and USER LEVEL DATA–II in Table 3, and observe a consistent reduction as the graph is incrementally refined.

From an uncertainty modeling perspective, our histogram-based formulation is closely related to recent work on semantic entropy in language models (Farquhar et al., 2024; Nikitin et al., 2024; Kuhn et al., 2023). However, in our setting, NLPSCM restricts the LM output space to the finite set of PAG edge types $\mathbf{E}^{\text{PAG}}$, yielding purely categorical predictions. Consequently, semantic entropy can be computed directly as the Shannon entropy of empirical histograms over sampled edge predictions, without requiring embedding-based clustering. While conceptually aligned with prior formulations of semantic entropy, this arises naturally in our setting from the discrete decision space induced by PAG edge selection.

## 4.3 LM Interaction: Sequential Optimization

Given the stochastic LM responses and cumulative histogram-based estimates of edge uncertainty, we next address how to allocate LM queries efficiently across candidate edges. We model LM interactions $f_{\text{LM}}$ as a *sequential optimization* problem under a limited budget. At each batch $i$, up to $m^{\text{L}}$ calls to the LM are allowed. The objective is to strategically allocate these calls to refine the edge distribution $\mathcal{H}^{\mathbf{E}_i}$ and expand the set of background knowledge $\mathcal{B}_i$.

In the edge refinement setting, each query corresponds to selecting a candidate edge $e$ and querying $f_{\text{LM}}$ to reduce uncertainty about its type. The LM's response is treated as a noisy sample from the underlying distribution over edge types. This induces a natural trade-off: we must *explore* uncertain edges to improve estimates and *exploit* promising edges that are likely to yield useful and increasing background knowledge.

**Algorithm 1** LM-augmented structure learning

**Require:** $\mathbf{D}_i, \mathcal{H}^{\mathbf{E}(i-1)}, \mathcal{H}^{\mathbf{L}(i-1)}, \mathcal{B}_{(i-1)}, m^{\mathbf{E}}, m^{\mathbf{L}},$
**Ensure:** $\mathcal{H}^{\mathbf{E}_i}, \mathcal{H}^{\mathbf{L}_i}, \mathcal{B}_i$
1:  **Initial causal structure**
2:  $f_{\mathrm{CD}} : \mathbf{D}_i \times \mathcal{B}_{(i-1)} \to \mathcal{G}^{\mathrm{D_i}}$
3:  **Expert-guided causal structure refinement**

4:  $f_{\mathrm{LM}} : \mathcal{G}^{\mathrm{D_i}} \times \mathcal{H}^{\mathbf{E}(i-1)} \times \mathcal{B}_{(i-1)} \times m^{\mathbf{E}} \to (\mathcal{H}^{\mathbf{E}_i}, \mathcal{B}_i)$
5:  **Expert-suggested latent confounder**
6:  **for** $A \leftrightarrow B$ in $\mathcal{B}_i$ **do**
7:      $f_{\mathrm{LM}} : \mathcal{H}^{\mathbf{E}_i} \times \mathcal{H}^{\mathbf{L}(i-1)} \times A \times B \times m^{\mathbf{L}} \to \mathcal{H}^{\mathbf{L}_i}$
8:  **end for**
9:  **return** $\mathcal{H}^{\mathbf{E}_i}, \mathcal{H}^{\mathbf{L}_i}, \mathcal{B}_i$

**Notation:**
$m^{\mathbf{E}}, m^{\mathbf{L}}$: LM budget for edges and confounder
$\mathcal{H}^{\mathbf{E}}, \mathcal{H}^{\mathbf{L}}$: histograms for edges and confounders
$\mathcal{I}^{\mathrm{P}}, \mathcal{I}^{\rho}$: Prompt for prior and corelation

**Algorithm 2** Bayesian Parameter Estimation

**Require:** $\mathbf{D}_i, \eta, \mathcal{G}^{\mathrm{X_i}}, \mathcal{I}^{\mathrm{P}}, \mathcal{I}^{\rho}$
**Ensure:** $\phi$
1:  **Initialization**
2:  Warm-start for $\phi^{\mathrm{O}}$
       $\phi^{\mathrm{O}} \in \arg\max_{\phi^{\mathrm{O}}} p(\mathbf{D}_i \,|\, \mathcal{G}^{\mathrm{X_i}}_{-\mathbf{V}^{\mathrm{L}}}, \phi^{\mathrm{O}})$
3:  Get prior over $\mathbf{V}^{\mathrm{L}}$
       $f_{\mathrm{LM}} : \mathcal{G}^{\mathrm{X_i}} \times \mathbf{V}^{\mathrm{L}} \times \mathcal{I}^{\mathrm{P}} \to \mathcal{N}(\boldsymbol{m}_p, \boldsymbol{S}_p)$
4:  Get correlation $\rho(\mathbf{V}^{\mathrm{L}}, \mathbf{V}^{\mathrm{O}})$
       $f_{\mathrm{LM}} : \mathcal{G}^{\mathrm{X_i}} \times \mathcal{I}^{\rho} \to \rho(\mathbf{V}^{\mathrm{L}}, \mathbf{V}^{\mathrm{O}})$
5:  Initialize $\boldsymbol{\theta}^{\mathrm{L}}$ to $\rho(\mathbf{V}^{\mathrm{L}}, \mathbf{V}^{\mathrm{O}})$ or randomly
6:  **Iterative Optimization**
7:  **while** not converged **do**
8:      Compute posterior over $\mathbf{V}^{\mathrm{L}}$ using Eq. (10)
9:      Optimize $\phi$ variables using Eq. (11)
10: **end while**
11: **return** $\phi$

Formally, we cast LM interactions as a sequential decision-making problem:

$$\textbf{Arms:} \quad \mathcal{A} = \{\text{All possible edges between variables, } \mathbf{E}^{\mathrm{PAG}}\}, \tag{6}$$

$$\textbf{Reward:} \quad r_k(e) = \text{Information gain from querying edge } e \text{ at step } k, \tag{7}$$

$$\textbf{Policy:} \quad \pi : \mathcal{H}^{\mathbf{E}_i} \times \mathcal{G}^{\mathrm{D_i}} \to e \quad \text{(Edge selection rule).} \tag{8}$$

The optimization objective is to maximize cumulative information gain over $m^{\mathrm{L}}$ LM calls, balancing both the expansion of background knowledge, and the uncertainty reduction in $\mathcal{H}^{\mathbf{E}_i}$. To implement $\pi$, since the true information gain $r_k(e)$ is not known before querying, we propose a scoring function that serves as a proxy for the expected reward, jointly accounting for epistemic uncertainty, proximity to background knowledge thresholds, and exploration,

$$S_i^e = w_1 E_i^e + w_2 \left( \frac{1}{TD_i^e} \right) + w_3 \sqrt{\frac{\log T_i}{T_i^e}}, \quad \text{s.t.} \quad TD_i^e = \tau_i - \max(\mathcal{H}^{\mathbf{E}_i}(e)), \tag{9}$$

where $TD_i^e$ is the threshold distance from being included in background knowledge, $E_i^e, T_i, T_i^e$ are as defined in Eq. (5), and $w_1, w_2, w_3$ are hyper-parameters controlling the trade-off. After querying, the LM's response updates the histogram $\mathcal{H}^{\mathbf{E}_i}(e)$, which constitutes the realized information gain. At each step, the edge $e^* = \arg\max_e S_i^e$ is selected for LM interaction. Fig. 4 showcases the effectiveness of the proposed selection score (*cf.* Eq. (9)), with details discussed in Sec. 6.1. This formulation generalizes to other expert-guided tasks (*e.g.,* confounder detection) by redefining the arms and reward function.

While the connection to multi-armed bandits and the derivation of regret bounds are appealing in this setup, LM's stochastic nature, inter-dependent arms, and implicit priors induced by causal structure constraints warrant caution. We discuss this in detail in Sec. D.

## 5 Bayesian Parameter Estimation

Once we obtain a causal structure $\mathcal{G}^{\mathrm{X_i}}$, we address the critical task of *parameter estimation* within the Structural Equation Model (SEM). The parameters $\phi = \{\boldsymbol{\theta}, \sigma^2\}$ include edge weights (coefficients) and noise parameters. With $\mathcal{G}^{\mathrm{X_i}}$ potentially containing both observed and latent variables $\mathbf{V}^{\mathrm{X_i}} = \{\mathbf{V}^{\mathrm{O}}, \mathbf{V}^{\mathrm{L}}\}$, we represent observed variable edges as $\boldsymbol{\theta}^{\mathrm{O}}$ and latent confounder edges as $\boldsymbol{\theta}^{\mathrm{L}}$, giving $\boldsymbol{\theta} = \{\boldsymbol{\theta}^{\mathrm{O}}, \boldsymbol{\theta}^{\mathrm{L}}\}$.

Table 4: **NLPSCM improves causal discovery:** We experiment with six datasets–number of observed variables ranges 5 (small) to 37 (large)– using two paradigms: *Only-Data* and *Data-LM*. We evaluate with 5 metrics: *Modified SHD, SID, Precision, Recall, F1.* All methods use GPT-3.5$_{\text{turbo}}$ as an LM with temperature 1. We report mean and standard deviation over 5 runs and perform significance test with $\alpha = 0.05$.

| Dataset | Approach | Method | Mod. SHD ↓ | SID ↓ | Precision ↑ | Recall ↑ | F1 Score ↑ |
|---|---|---|---|---|---|---|---|
| EARTHQUAKE ($d=5$) | Only-Data | FCI-Cumulative | $2.00_{\pm 0.00}$ | $(0.00, 5.00)_{\pm(0.00, 0.00)}$ | $\mathbf{1.00}_{\pm 0.00}$ | $0.50_{\pm 0.00}$ | $0.67_{\pm 0.00}$ |
| | | FCI-Vanilla | $3.60_{\pm 0.80}$ | $(8.20, 9.20)_{\pm(3.60, 1.60)}$ | $0.20_{\pm 0.40}$ | $0.05_{\pm 0.10}$ | $0.08_{\pm 0.16}$ |
| | | FCI-Iterative | $5.00_{\pm 1.67}$ | $(12.20, 12.20)_{\pm(4.66, 4.66)}$ | $0.30_{\pm 0.27}$ | $0.20_{\pm 0.19}$ | $0.24_{\pm 0.22}$ |
| | | FCI-Heuristics | $3.60_{\pm 0.80}$ | $(8.20, 9.20)_{\pm(3.60, 1.60)}$ | $0.20_{\pm 0.40}$ | $0.05_{\pm 0.10}$ | $0.08_{\pm 0.16}$ |
| | Data-LM | LLM-first | $6.00_{\pm 0.82}$ | $(15.00, 15.00)_{\pm(0.82, 0.82)}$ | $0.11_{\pm 0.16}$ | $0.08_{\pm 0.12}$ | $0.09_{\pm 0.13}$ |
| | | ILS-CSL | $2.38_{\pm 0.96}$ | $(5.25, 6.50)_{\pm(0.83, 2.69)}$ | $\mathbf{0.88}_{\pm 0.22}$ | $0.44_{\pm 0.21}$ | $0.56_{\pm 0.21}$ |
| | | NLPSCM | $\mathbf{1.00}_{\pm 0.63}$ | $\mathbf{(2.20, 2.20)}_{\pm(1.60, 1.60)}$ | $\mathbf{1.00}_{\pm 0.00}$ | $\mathbf{0.75}_{\pm 0.16}$ | $\mathbf{0.85}_{\pm 0.11}$ |
| ASIA ($d=8$) | Only-Data | FCI-Cumulative | $7.00_{\pm 0.00}$ | $(23.00, 49.00)_{\pm(0.00, 0.00)}$ | $0.00_{\pm 0.00}$ | $0.00_{\pm 0.00}$ | $0.00_{\pm 0.00}$ |
| | | FCI-Vanilla | $7.80_{\pm 0.75}$ | $(30.00, 35.00)_{\pm(5.90, 2.45)}$ | $0.00_{\pm 0.00}$ | $0.00_{\pm 0.00}$ | $0.00_{\pm 0.00}$ |
| | | FCI-Iterative | $8.00_{\pm 1.26}$ | $(33.00, 35.00)_{\pm(7.46, 7.46)}$ | $0.45_{\pm 0.24}$ | $0.23_{\pm 0.15}$ | $0.29_{\pm 0.18}$ |
| | | FCI-Heuristics | $7.80_{\pm 0.75}$ | $(30.00, 35.00)_{\pm(5.90, 2.45)}$ | $0.00_{\pm 0.00}$ | $0.00_{\pm 0.00}$ | $0.00_{\pm 0.00}$ |
| | Data-LM | LLM-first | $7.33_{\pm 0.94}$ | $(27.67, 27.67)_{\pm(2.49, 2.49)}$ | $0.58_{\pm 0.12}$ | $0.29_{\pm 0.06}$ | $0.39_{\pm 0.08}$ |
| | | ILS-CSL | $6.50_{\pm 0.50}$ | $(28.50, 28.50)_{\pm(3.20, 3.20)}$ | $\mathbf{0.79}_{\pm 0.12}$ | $0.28_{\pm 0.11}$ | $0.40_{\pm 0.12}$ |
| | | NLPSCM | $\mathbf{4.60}_{\pm 1.02}$ | $\mathbf{(13.60, 13.60)}_{\pm(3.83, 3.83)}$ | $\mathbf{0.80}_{\pm 0.12}$ | $\mathbf{0.60}_{\pm 0.12}$ | $\mathbf{0.67}_{\pm 0.08}$ |
| USER LEVEL DATA-I ($d=9$) | Only-Data | FCI-Cumulative | $15.00_{\pm 2.77}$ | $(47.80, 47.80)_{\pm(4.62, 4.62)}$ | $0.72_{\pm 0.18}$ | $0.37_{\pm 0.09}$ | $0.47_{\pm 0.09}$ |
| | | FCI-Vanilla | $21.30_{\pm 3.50}$ | $(61.60, 61.60)_{\pm(5.54, 5.54)}$ | $0.41_{\pm 0.18}$ | $0.22_{\pm 0.10}$ | $0.29_{\pm 0.13}$ |
| | | FCI-Iterative | $\mathbf{5.60}_{\pm 1.20}$ | $(23.40, 23.40)_{\pm(7.94, 7.94)}$ | $\mathbf{0.93}_{\pm 0.04}$ | $0.77_{\pm 0.02}$ | $\mathbf{0.84}_{\pm 0.03}$ |
| | | FCI-Heuristics | $21.30_{\pm 3.50}$ | $(61.60, 61.60)_{\pm(5.54, 5.54)}$ | $0.41_{\pm 0.18}$ | $0.22_{\pm 0.10}$ | $0.29_{\pm 0.13}$ |
| | Data-LM | LLM-first | $9.68_{\pm 1.67}$ | $(38.12, 38.12)_{\pm(4.06, 4.06)}$ | $0.80_{\pm 0.06}$ | $0.66_{\pm 0.04}$ | $0.72_{\pm 0.05}$ |
| | | ILS-CSL | $9.20_{\pm 1.41}$ | $(34.20, 34.20)_{\pm(3.49, 3.49)}$ | $0.82_{\pm 0.05}$ | $0.66_{\pm 0.05}$ | $0.73_{\pm 0.05}$ |
| | | NLPSCM | $\mathbf{5.00}_{\pm 0.71}$ | $\mathbf{(13.75, 13.75)}_{\pm(2.49, 2.49)}$ | $0.90_{\pm 0.04}$ | $\mathbf{0.83}_{\pm 0.02}$ | $\mathbf{0.86}_{\pm 0.02}$ |
| USER LEVEL DATA-II ($d=8$) | Only-Data | FCI-Cumulative | $19.80_{\pm 2.04}$ | $\mathbf{(40.00, 40.00)}_{\pm(3.03, 3.03)}$ | $0.15_{\pm 0.07}$ | $0.10_{\pm 0.04}$ | $0.12_{\pm 0.05}$ |
| | | FCI-Vanilla | $17.40_{\pm 1.83}$ | $\mathbf{(36.80, 39.40)}_{\pm(2.04, 3.38)}$ | $0.07_{\pm 0.13}$ | $0.02_{\pm 0.03}$ | $0.03_{\pm 0.05}$ |
| | | FCI-Iterative | $20.60_{\pm 1.77}$ | $(42.80, 43.40)_{\pm(4.07, 4.22)}$ | $0.06_{\pm 0.08}$ | $0.05_{\pm 0.07}$ | $0.05_{\pm 0.07}$ |
| | | FCI-Heuristics | $17.40_{\pm 1.83}$ | $\mathbf{(36.80, 39.40)}_{\pm(2.04, 3.38)}$ | $0.07_{\pm 0.13}$ | $0.02_{\pm 0.03}$ | $0.03_{\pm 0.05}$ |
| | Data-LM | LLM-first | $18.75_{\pm 0.43}$ | $(44.00, 44.00)_{\pm(1.00, 1.00)}$ | $0.18_{\pm 0.01}$ | $\mathbf{0.17}_{\pm 0.00}$ | $0.18_{\pm 0.00}$ |
| | | ILS-CSL | $\mathbf{16.90}_{\pm 1.50}$ | $(44.40, 47.80)_{\pm(6.05, 3.49)}$ | $0.16_{\pm 0.03}$ | $0.10_{\pm 0.04}$ | $0.12_{\pm 0.03}$ |
| | | NLPSCM | $\mathbf{16.33}_{\pm 1.80}$ | $(40.17, 40.17)_{\pm(3.14, 3.14)}$ | $\mathbf{0.26}_{\pm 0.04}$ | $0.18_{\pm 0.06}$ | $\mathbf{0.20}_{\pm 0.04}$ |
| CHILD ($d=19$) | Only-Data | FCI-Cumulative | $27.50_{\pm 0.00}$ | $\mathbf{(111.00, 131.00)}_{\pm(0.00, 0.00)}$ | $0.38_{\pm 0.00}$ | $0.36_{\pm 0.00}$ | $0.37_{\pm 0.00}$ |
| | | FCI-Vanilla | $28.00_{\pm 1.48}$ | $(129.20, 133.20)_{\pm(10.46, 10.76)}$ | $0.38_{\pm 0.04}$ | $0.26_{\pm 0.05}$ | $0.31_{\pm 0.04}$ |
| | | FCI-Iterative | $32.10_{\pm 1.16}$ | $(149.00, 164.40)_{\pm(7.16, 10.33)}$ | $0.27_{\pm 0.03}$ | $0.26_{\pm 0.04}$ | $0.26_{\pm 0.03}$ |
| | | FCI-Heuristics | $28.00_{\pm 1.48}$ | $(129.20, 133.20)_{\pm(10.46, 10.76)}$ | $0.38_{\pm 0.04}$ | $0.26_{\pm 0.05}$ | $0.31_{\pm 0.04}$ |
| | Data - LLM | LLM-first | $31.67_{\pm 2.05}$ | $(172.00, 172.00)_{\pm(13.42, 13.42)}$ | $0.29_{\pm 0.05}$ | $0.30_{\pm 0.05}$ | $0.30_{\pm 0.05}$ |
| | | ILS-CSL | $32.00_{\pm 2.10}$ | $(154.00, 154.00)_{\pm(26.53, 26.53)}$ | $0.28_{\pm 0.05}$ | $0.28_{\pm 0.05}$ | $0.28_{\pm 0.05}$ |
| | | NLPSCM | $\mathbf{25.50}_{\pm 0.89}$ | $\mathbf{(103.40, 112.20)}_{\pm(8.09, 7.05)}$ | $\mathbf{0.43}_{\pm 0.01}$ | $\mathbf{0.44}_{\pm 0.02}$ | $\mathbf{0.43}_{\pm 0.01}$ |
| ALARM ($d=37$) | Only-Data | FCI-Cumulative | $\mathbf{45.00}_{\pm 0.00}$ | $\mathbf{(626.00, 626.00)}_{\pm(0.00, 0.00)}$ | $0.25_{\pm 0.00}$ | $0.02_{\pm 0.00}$ | $0.04_{\pm 0.00}$ |
| | | FCI-Vanilla | $49.50_{\pm 1.61}$ | $(617.80, 699.20)_{\pm(29.23, 68.43)}$ | $0.00_{\pm 0.00}$ | $0.00_{\pm 0.00}$ | $0.00_{\pm 0.00}$ |
| | | FCI-Iterative | $52.40_{\pm 6.21}$ | $(612.20, 636.80)_{\pm(49.87, 43.83)}$ | $0.33_{\pm 0.14}$ | $0.12_{\pm 0.05}$ | $0.17_{\pm 0.07}$ |
| | | FCI-Heuristics | $49.50_{\pm 1.61}$ | $(617.80, 699.20)_{\pm(29.23, 68.43)}$ | $0.00_{\pm 0.00}$ | $0.00_{\pm 0.00}$ | $0.00_{\pm 0.00}$ |
| | Data - LLM | LLM-first | $52.33_{\pm 3.09}$ | $(673.33, 673.33)_{\pm(25.63, 25.63)}$ | $\mathbf{0.33}_{\pm 0.08}$ | $0.13_{\pm 0.02}$ | $0.19_{\pm 0.03}$ |
| | | ILS-CSL | $51.20_{\pm 1.60}$ | $(657.20, 657.20)_{\pm(24.07, 24.07)}$ | $0.34_{\pm 0.05}$ | $0.10_{\pm 0.01}$ | $0.15_{\pm 0.02}$ |
| | | NLPSCM | $50.90_{\pm 1.32}$ | $\mathbf{(589.80, 591.00)}_{\pm(26.48, 25.59)}$ | $\mathbf{0.42}_{\pm 0.03}$ | $\mathbf{0.22}_{\pm 0.02}$ | $\mathbf{0.29}_{\pm 0.02}$ |

When no latent confounders exist ($\mathbf{V}^{\text{L}}=\emptyset$), standard Maximum Likelihood Estimation (MLE) optimizes the parameters $\boldsymbol{\phi}=\{\boldsymbol{\theta}^{\text{O}}, \sigma^2\}$ as $\boldsymbol{\phi} = \arg\max_{\boldsymbol{\phi}} \log p(\mathbf{D}_i \,|\, \mathcal{G}^{\text{X}_i}, \boldsymbol{\phi})$ using a conventional gradient-based methods. We focus on the more important and challenging scenario where latent confounders exist ($\mathbf{V}^{\text{L}}\neq\emptyset$).

With latent confounders, MLE is ill-posed and intractable. We instead employ an iterative Expectation–Maximization (EM) algorithm that incorporates LM-provided probability $p(\mathbf{V}^{\text{L}})$ and correlation $\rho(\mathbf{V}^{\text{O}}, \mathbf{V}^{\text{L}})$ about latent confounders. Specifically, we propose the following EM steps:

- **E-step:** Compute conditional posterior of latent confounder(s) given $\mathbf{D}_i$ and SEM parameters $\boldsymbol{\phi}$,

$$p(\mathbf{V}^{\text{L}} \,|\, \mathcal{G}^{\text{X}_i}, \mathbf{D}_i, \boldsymbol{\phi}) \propto p(\mathbf{D}_i \,|\, \mathcal{G}^{\text{X}_i}, \mathbf{V}^{\text{L}}, \boldsymbol{\phi})p(\mathbf{V}^{\text{L}}). \tag{10}$$

- **M-step:** Update parameters by maximizing the expected log-likelihood, incorporating LM-provided regularization for latent confounder edges,

$$\boldsymbol{\phi} \in \arg\max_{\boldsymbol{\phi}} \mathbf{E}_{p(\mathbf{V}^{\text{L}} \,|\, \mathcal{G}^{\text{X}_i}, \mathbf{D}_i, \boldsymbol{\phi})}[\log p(\mathbf{D}_i \,|\, \mathcal{G}^{\text{X}_i}, \mathbf{V}^{\text{L}}, \boldsymbol{\phi})] - \lambda \|\boldsymbol{\theta}^{\text{L}} - (\rho(\mathbf{V}^{\text{O}}, \mathbf{V}^{\text{L}})\sigma_{\mathbf{V}^{\text{O}}} \sigma_{\mathbf{V}^{\text{L}}}^{-1})\|_2. \tag{11}$$

Alg. 2 details the EM parameter estimation algorithm. In Sec. 6.3, we demonstrate the robustness and recovery capability of the proposed parameter estimation algorithm.

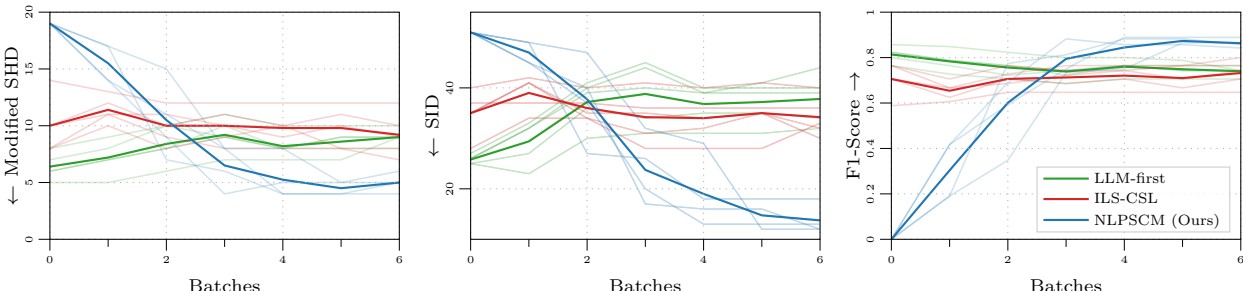

Figure 3: User Level Data - I: Performance evolution across batches for *Data-LM* methods. Left: Modified Structural Hamming Distance (↓), Middle: Structural Intervention Distance (↓), and Right: F1-Score (↑). NLPSCM consistently outperforms other approaches as data accumulation progresses.

## 6  Experiments

To provide a robust empirical examination, we conduct experiments on four language models: GPT-3.5$_{\text{turbo}}$, GPT4.1$_{\text{nano}}$, Llama3.1$_{\text{8B-instruct}}$, Qwen3$_{\text{4B-instruct}}$ and six datasets: Earthquake from Korb & Nicholson (2010), Asia from Lauritzen & Spiegelhalter (1988), User Level Data - I, User Level Data - II from Google & Kaggle (2018), Child from Spiegelhalter et al. (1993), and Alarm from Beinlich et al. (1989). They range in number of observed variables (nodes) from small (5) to medium (19) to large (37). Earthquake, Asia, Child, and Alarm datasets are standard benchmarks, providing observational data and ground-truth causal structures. To simulate a streaming batch setting, each dataset is split into batches. For the two User Level Data, which contain only observational data, the underlying DAG is inferred using DirectLiNGAM, an algorithm proposed by Shimizu et al. (2011). We treat the DAG inferred from DirectLiNGAM as the ground truth for evaluation. Further details on the data sets and simulation process are provided in Sec. A.

Our evaluation metrics include a *modified* Structural Hamming Distance (Mod. SHD), which extends SHD to account for uncertain edges in PAG; Structural Intervention Distance (SID); and precision, recall, and F1-score for causal relations that are certain. Together, these metrics assess structural accuracy, interventional soundness, and edge-wise discovery performance (details in Sec. C).

### 6.1  Structure Learning

We evaluate NLPSCM against *Only-Data* and *Data-LM* baselines in Table 4. As shown in Table 1, *Only-LM* methods exhibit *overly optimistic* behavior, producing globally plausible but locally unreliable causal structures. This highlights the need for data-grounded post-processing. Table 4 compares NLPSCM with several baselines, including multiple FCI variants (cumulative, vanilla, iterative, heuristics), as well as *Data-LM* approaches (LM-first, and ILS-CSL proposed by Ban et al. (2023a)). Across all evaluation metrics, NLPSCM consistently outperforms baselines, which also holds for different LM temperatures (Table A3). While Table 4 reports metrics for the final batch, we also show performance evolution across batches, a crucial step in sequential settings (*cf.* Fig. 3). We justify the use of FCI over other causal discovery algorithms in Sec. B. We provide more experiment details in Sec. F.

Finally, going beyond GPT-3.5$_{\text{turbo}}$ (Table 4), results with recent LMs, GPT-4o and GPT-5 (Table A2) show good performance gains for NLPSCM across LMs. The much higher inference cost of GPT-4o and GPT-5, over GPT-3.5$_{\text{turbo}}$ constrain their use for large set of experiments.

**Structure learning ablations**  We ablate two key components of NLPSCM: *(i)* selection score for sequential optimization, and *(ii)* dynamic background threshold. Results of ablations performed on User Level Data - I are shown in Fig. 4.

Effectiveness of proposed selection score (Eq. (9)) in guiding edge selection under a fixed budget of LM queries, is compared against a random-selection baseline. Fig. 4 (Left, Middle) shows that NLPSCM achieves significantly better Mod. SHD and F1-score across batches, demonstrating the benefit of a principled edge selection policy in sequential structure learning.

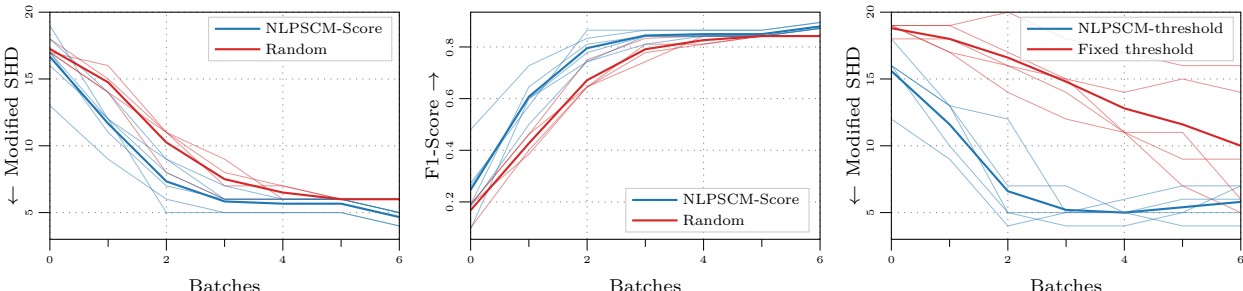

Figure 4: **Structure learning ablation:** The impact of two key components: *selection score* and *dynamic background threshold*. (Left, Middle) Modified SHD ($\downarrow$) and F1-score ($\uparrow$) on the USER LEVEL DATA - I dataset, comparing NLPSCM—*selection score* against random selection. (Right) Modified SHD($\downarrow$) comparing NLPSCM— *dynamic threshold* with a conventional fixed threshold.

We also assess the impact of dynamic background threshold (Eq. (5)) used to promote edges from the histogram $\mathcal{H}^{\mathbf{E}}$ into the background knowledge $\mathcal{B}$. Fig. 4 (Right) reports Mod. SHD over batches, highlighting the advantages of a dynamic threshold over a conventional fixed one. The adaptive mechanism yields more stable and accurate graph recovery throughout the learning process.

## 6.2 Other LM Families and Memorization in LMs

We evaluate NLPSCM using three *recent* LMs across different families: Llama3.1$_{8B\text{-instruct}}$, Qwen3$_{4B\text{-instruct}}$, and GPT4.1$_{nano}$. The first two are open models. We evaluate on two datasets: CHILD and the real-world USER LEVEL DATA - I from Google & Kaggle (2018). Measuring performance on the 5 metrics over 5 runs, the results in Table 5 are consistent with those of the previously presented GPT3.5$_{Turbo}$ model, demonstrating robustness across LMs with different training data, and architectures.

We introduce an additional baseline which does causal discovery using LM and Pearson correlation in the prompt, namely BFS, as proposed by Jiralerspong et al. (2024). A comparison of NLPSCM with BFS also serves an important purpose by throwing light on the issue of potential memorization of datasets seen in training by LMs. As noted by Jiralerspong et al. (2024), their method relies on the LM's training knowledge (Sec. 5 of their paper), thereby recognizing the dependence on memorization. To make a fair comparison, we use a recent model GPT4.1$_{nano}$. Results reported in the bottom panel of Table 5 show that for both CHILD and USER LEVEL DATA - I, NLPSCM beats BFS handily. To elucidate about memorization, we compare the difference-of-differences in metric-values between BFS and NLPSCM. The difference is *smaller* for CHILD dataset, a standard causal discovery benchmark that is likely to be a part of the LM training data. In contrast, the difference is *much larger* for the USER LEVEL DATA-I dataset. The latter dataset or its causal graph is unlikely to have been seen by any LM due to the processing and construction of attributes we performed in this dataset, from the publicly available large data at Google & Kaggle (2018). That BFS performs worse highlights its reliance on memorized knowledge, whereas NLPSCM demonstrates greater robustness across datasets.

## 6.3 Parameter Estimation

The datasets used in Sec. 6.1 do not contain latent confounders in their causal structures. Consequently, parameter estimation reduces to standard maximum likelihood estimation, which NLPSCM (Sec. 5) replicates by design. To meaningfully evaluate NLPSCM in the presence of latent confounders, we consider a real-world dataset: the RED-WINE QUALITY from Cortez et al. (2009) (details in Sec. A).

For the RED-WINE QUALITY, NLPSCM's structure learning algorithm predicts a latent confounder between variables *quality* and *density*. Applying DirectLiNGAM on the full dataset indicates that the true confounder is *alcohol content*, aligning with domain knowledge. Following Sec. 4.2, we query an LM to identify potential latent confounders using world knowledge. Fig. 6 shows a histogram of LM predictions, with *alcohol_content* emerging as the top candidate. We incorporate this LM-provided confounder into NLPSCM's parameter estimation pipeline and query the LM for its marginal distribution. Table 6 presents the LM-provided

Table 5: **NLPSCM improves causal discovery across language model families:** We showcase results on the CHILD and USER LEVEL DATA-I datasets using three language models: *Llama3.1$_{8B\text{-}instruct}$*, *Qwen3$_{4B\text{-}instruct}$* and a GPT-4.1$_{nano}$. We report 5 metrics: *Modified SHD, SID, Precision, Recall, F1*, with mean and standard deviation over 5 runs and perform significance test with $\alpha = 0.05$.

| Dataset | Model | Method | Mod. SHD ↓ | SID ↓ | Precision ↑ | Recall ↑ | F1 Score ↑ |
|---|---|---|---|---|---|---|---|
| CHILD | Llama3.1$_{8B\text{-}instruct}$ | LLM-first | $45.80_{\pm 1.94}$ | $(211.20, 211.20)_{\pm (15.25,\ 15.25)}$ | $0.13_{\pm 0.01}$ | $0.19_{\pm 0.02}$ | $0.16_{\pm 0.01}$ |
| | | ILS-CSL | $27.60_{\pm 2.24}$ | $(146.40, 146.40)_{\pm (15.93,\ 15.93)}$ | $0.37_{\pm 0.05}$ | $0.38_{\pm 0.07}$ | $0.38_{\pm 0.06}$ |
| | | NLPSCM | $\mathbf{24.60}_{\pm 0.80}$ | $\mathbf{(101.80, 101.80)}_{\pm (8.11,\ 8.11)}$ | $\mathbf{0.44}_{\pm 0.01}$ | $\mathbf{0.45}_{\pm 0.02}$ | $\mathbf{0.44}_{\pm 0.01}$ |
| | Qwen3$_{4B\text{-}instruct}$ | LLM-first | $35.80_{\pm 2.04}$ | $(173.80, 173.80)_{\pm (14.13,\ 14.13)}$ | $0.27_{\pm 0.02}$ | $\mathbf{0.42}_{\pm 0.02}$ | $0.34_{\pm 0.01}$ |
| | | ILS-CSL | $30.60_{\pm 2.42}$ | $(196.80, 196.80)_{\pm (14.13,\ 14.13)}$ | $0.30_{\pm 0.06}$ | $0.28_{\pm 0.05}$ | $0.29_{\pm 0.05}$ |
| | | NLPSCM | $\mathbf{21.20}_{\pm 0.75}$ | $\mathbf{(107.80, 107.80)}_{\pm (13.42,\ 13.42)}$ | $\mathbf{0.51}_{\pm 0.01}$ | $0.32_{\pm 0.03}$ | $\mathbf{0.39}_{\pm 0.02}$ |
| | GPT4.1$_{nano}$ | LLM-first | $31.83_{\pm 2.61}$ | $(203.50, 203.50)_{\pm (23.06,\ 23.06)}$ | $0.16_{\pm 0.01}$ | $0.14_{\pm 0.03}$ | $0.15_{\pm 0.02}$ |
| | | ILS-CSL | $31.60_{\pm 2.06}$ | $\mathbf{(147.40, 147.40)}_{\pm (11.25,\ 11.25)}$ | $0.29_{\pm 0.05}$ | $0.19_{\pm 0.04}$ | $0.23_{\pm 0.03}$ |
| | | BFS | $34.75_{\pm 7.80}$ | $(252.50, 252.50)_{\pm (23.36,\ 23.36)}$ | $0.18_{\pm 0.08}$ | $0.12_{\pm 0.04}$ | $0.14_{\pm 0.03}$ |
| | | BFS$_{corr}$ | $32.30_{\pm 4.16}$ | $(301.10, 301.10)_{\pm (14.42,\ 14.42)}$ | $0.21_{\pm 0.03}$ | $0.15_{\pm 0.01}$ | $0.18_{\pm 0.01}$ |
| | | NLPSCM | $\mathbf{27.90}_{\pm 1.61}$ | $\mathbf{(143.50, 165.50)}_{\pm (12.23,\ 35.97)}$ | $\mathbf{0.42}_{\pm 0.06}$ | $\mathbf{0.23}_{\pm 0.04}$ | $\mathbf{0.30}_{\pm 0.03}$ |
| USER LEVEL DATA-I | Llama3.1$_{8B\text{-}instruct}$ | LLM-first | $11.50_{\pm 0.50}$ | $(42.50, 42.50)_{\pm (1.50,\ 1.50)}$ | $0.74_{\pm 0.01}$ | $0.61_{\pm 0.03}$ | $0.67_{\pm 0.02}$ |
| | | ILS-CSL | $9.75_{\pm 1.64}$ | $(32.25, 32.25)_{\pm (3.56,\ 3.56)}$ | $0.78_{\pm 0.05}$ | $0.66_{\pm 0.03}$ | $0.72_{\pm 0.04}$ |
| | | NLPSCM | $\mathbf{7.50}_{\pm 0.45}$ | $\mathbf{(25.80, 25.40)}_{\pm (2.32,\ 4.22)}$ | $\mathbf{0.92}_{\pm 0.07}$ | $\mathbf{0.69}_{\pm 0.01}$ | $\mathbf{0.79}_{\pm 0.01}$ |
| | Qwen3$_{4B\text{-}instruct}$ | LLM-first | $23.67_{\pm 1.11}$ | $(64.33, 64.33)_{\pm (0.94,\ 0.94)}$ | $0.35_{\pm 0.04}$ | $0.28_{\pm 0.04}$ | $0.31_{\pm 0.04}$ |
| | | ILS-CSL | $20.60_{\pm 4.18}$ | $(56.80, 56.80)_{\pm (7.36,\ 7.36)}$ | $0.45_{\pm 0.13}$ | $0.36_{\pm 0.10}$ | $0.40_{\pm 0.12}$ |
| | | NLPSCM | $\mathbf{9.80}_{\pm 2.04}$ | $\mathbf{(26.60, 26.60)}_{\pm (6.83,\ 6.83)}$ | $\mathbf{0.91}_{\pm 0.08}$ | $\mathbf{0.54}_{\pm 0.08}$ | $\mathbf{0.67}_{\pm 0.07}$ |
| | GPT4.1$_{nano}$ | LLM-first | $19.42_{\pm 1.14}$ | $(72.10, 72.10)_{\pm (5.36,\ 5.36)}$ | $0.76_{\pm 0.07}$ | $0.32_{\pm 0.03}$ | $0.45_{\pm 0.03}$ |
| | | ILS-CSL | $18.00_{\pm 1.41}$ | $(54.00, 54.00)_{\pm (2.53,\ 2.53)}$ | $0.53_{\pm 0.05}$ | $\mathbf{0.42}_{\pm 0.03}$ | $0.47_{\pm 0.04}$ |
| | | BFS | $36.60_{\pm 8.96}$ | $(65.80, 65.80)_{\pm (2.17,\ 2.17)}$ | $0.13_{\pm 0.06}$ | $0.29_{\pm 0.14}$ | $0.18_{\pm 0.06}$ |
| | | BFS$_{corr}$ | $24.00_{\pm 6.24}$ | $(42.20, 42.20)_{\pm (6.53,\ 6.53)}$ | $0.17_{\pm 0.12}$ | $0.28_{\pm 0.11}$ | $0.21_{\pm 0.11}$ |
| | | NLPSCM | $\mathbf{10.60}_{\pm 2.13}$ | $\mathbf{(34.80, 35.20)}_{\pm (5.56,\ 5.84)}$ | $\mathbf{0.97}_{\pm 0.04}$ | $\mathbf{0.43}_{\pm 0.06}$ | $\mathbf{0.60}_{\pm 0.06}$ |

Gaussian distributions. Notably, when queried with obscure variables (*e.g.,* (cat, mouse)), LM often defaults to unit Gaussian $\mathcal{N}(0, 1)$.

To assess parameter recovery, we track evolution of $\boldsymbol{\theta}$ over sequential batches. As a performance metric, we compute the $\ell_2$-norm error $\|\boldsymbol{\theta}^\star - \boldsymbol{\theta}\|_2$, where $\boldsymbol{\theta}^\star$ denotes the parameters obtained via MLE assuming the confounder (*alcohol_content*) is observed. Fig. 5 visualizes this convergence behavior.

**Parameter estimation: robustness and recovery** We demonstrate robustness of the proposed parameter estimation algorithm under misspecified or ill-informed priors. Based on domain knowledge and observational data, we estimate latent confounder *alcohol_content* to follow distribution $\mathcal{N}(11, 1.0)$. To stress-test NLPSCM, we experiment with three alternative priors: $\mathcal{N}(12.5, 2.5)$ (suggested by GPT-3.5$_{turbo}$), $\mathcal{N}(0, 1)$, and a severely misspecified prior $\mathcal{N}(50, 1.5)$. Fig. 5 illustrates the evolution of the learned parameters $\boldsymbol{\theta}$ across training batches for each prior. As expected and aligned with the Bayesian principle, convergence is slower when initialized with an inaccurate prior. Nevertheless, the model progressively refines its estimates as more data is processed, ultimately converging towards $\boldsymbol{\theta}^\star$. This demonstrates both the robustness and recovery capabilities of NLPSCM's Bayesian parameter estimation algorithm—even under poor initialization. Additionally, we incorporate LM-suggested Pearson correlation coefficients in the *M-step* objective to further guide estimation, Eq. (11) (see discussion and results in Sec. E.).

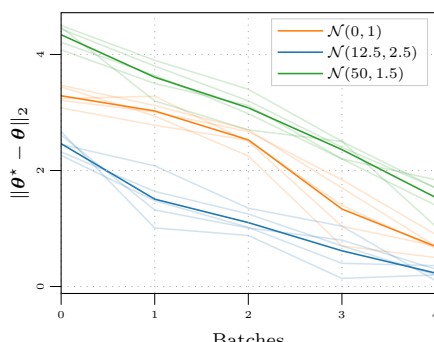

Figure 5: **Parameter Estimation:** Convergence of parameters and robustness to prior misspecification as more batches are processed.

# 7 Discussion and Conclusion

We present NLPSCM (Noisy Language Prior in Sequential Causal Modeling)—a Bayesian-inspired framework for causal structure discovery and parameter (edge weights) estimation in sequential, batch-wise data settings. By treating language models (LMs) as noisy surrogate experts, NLPSCM addresses the dual *LM-induced* and *data-induced* biases. A key contribution is the representation shift from DAGs to PAGs, allowing uncertainty and confounders to be modeled explicitly. Through LM interactions modeled in sequential optimization

Table 6: **LM-predicted priors for confounding variable:** LM suggest relevant Gaussian priors when the confounder is meaningful and default to $\mathcal{N}(0, 1)$ when uncertain.

| Variables | GPT-3.5$_{\text{turbo}}$ | GPT-4o |
|---|---|---|
| density← alcohol → quality | $\mathcal{N}(12.5, 2.5)$ | $\mathcal{N}(10.5, 1.2)$ |
| density ← alcohol → volatile-acidity | $\mathcal{N}(12.5, 2.5)$ | $\mathcal{N}(10.5, 1.2)$ |
| cat ← alcohol → mouse | $\mathcal{N}(0, 1)$ | $\mathcal{N}(0, 1)$ |
| bed ← alcohol → shopping | $\mathcal{N}(0, 1)$ | $\mathcal{N}(0, 1)$ |

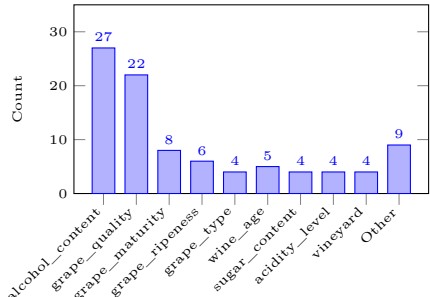

Figure 6: LM predicted confounders.

framework and EM style iterative parameter estimation algorithm, NLPSCM improves both structural accuracy and parameter recovery in hybrid *data-LM* pipelines. NLPSCM leverages global LM knowledge while staying grounded in local data, and offers a robust foundation for hybrid causal discovery in the presence of sequential, batched data.

**Limitations and Future Work**   In the Bayesian formulation in NLPSCM, incorporating Dirichlet or hierarchical Bayesian priors over LM judgments could provide another way to capture LM uncertainty. Extending to fully probabilistic inference over graph structures, rather than edge-level belief tracking, is another promising direction. Future work may also explore adaptive calibration of LM responses or memory-based accumulation of observational data across batches, following Chang et al. (2023). We employ FCI for causal discovery due to its compatibility with our setup (Section 3). The current framework assumes that each latent confounder may affect two observed variables; extending to multi-variable confounding is a direction for future work. Systematically evaluating the robustness of NLPSCM for other causal discovery algorithms remains important future work. Additionally, extending our approach to incorporate interventional data and active learning strategies could improve sample efficiency and the quality of discovered causal structures. Systematic evaluation of LM accuracy in confounder identification across diverse domains also remains a useful future effort. Finally, a theoretical analysis of sequential optimization and a bandit-style framework remains an open problem, discussed preliminarily in Sec. D.

**Broader Impact**   NLPSCM combines observational data with LM-derived knowledge to discover causal structures, which may inform downstream decisions in domains such as healthcare, policy, and business. As with any causal discovery method, the learned structures reflect the assumptions and limitations of the underlying algorithms, data quality, and in our case, the reliability of LM priors. We recommend that practitioners treat discovered structures as hypotheses to be validated through domain expertise or interventional studies before acting on them.

## Acknowledgments

This work was primarily conducted while PV was an intern at Adobe Research, Bangalore. PV and AS acknowledge funding from the Research Council of Finland (grants 362408, 339730). We acknowledge CSC-IT Center for Science, Finland, and the Aalto Science-IT project for the computational resources.

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
