## Appendix

We organize the appendix as follows: Sec. A provides detailed descriptions of all datasets used and simulation details; Sec. B discusses the choice of FCI over other causal discovery algorithms; Sec. C outlines the evaluation metrics employed to assess method performance; Sec. D discusses the connection between the proposed sequential optimization approach and the multi-armed bandit framework; Sec. E describes the parameter estimation algorithm and its robustness to LM-derived information; and Sec. F presents the experimental setup along with additional implementation and hyperparameter details.

## A   Dataset and Simulation Details

EARTHQUAKE   The EARTHQUAKE dataset is a widely used synthetic benchmark in causal discovery. It comprises a small, acyclic, and semantically meaningful graph over five binary variables: *Burglary, Earthquake, Alarm, JohnCalls,* and *MaryCalls.* In this structure, an earthquake or burglary can trigger the alarm, which in turn influences whether John or Mary calls. The ground-truth causal graph is depicted in Fig. A1a. For our experiments, we aggregate data from six CSV files (`earthquake_250_1-6.csv`) available at `https://github.com/andrewli77/MINOBS-anc/`, resulting in a combined dataset of 1500 samples. As the dataset contains Boolean values (Yes/No) we convert them to a numerical format (1/0) producing the full dataset $\mathbb{R}^{(1500,5)}$.

ASIA   The ASIA dataset is another canonical benchmark in causal discovery and probabilistic inference. It consists of eight binary variables (*e.g., VisitToAsia, Tuberculosis, Smoking, LungCancer*) structured in a directed acyclic graph (DAG) with known semantics, as shown in Fig. A1b. We construct the dataset by merging all samples from the six CSV files (`asia_250_1-6.csv`) provided at `https://github.com/andrewli77/MINOBS-anc/`, yielding a total of 1500 samples. As with the EARTHQUAKE dataset, all boolean values are converted to numerical format producing the full dataset $\mathbb{R}^{(1500,8)}$.

USER LEVEL DATA - I   The USER LEVEL DATA - I dataset is derived from the publicly available Google Analytics Sample Dataset, accessible via Google BigQuery (detailed in (Google & Kaggle, 2018)). It captures real-world e-commerce interactions from the Google Merchandise Store. We focus on a curated subset of user-level features that reflect engagement, browsing behavior, and purchase intent. The selected variables include *Proximity to Transaction, Number of Add To Cart, Number of Product Clicks, Number of Sessions on iOS, Number of Promo Hits (Android), Number of Cheap Products Viewed, Number of Page Hits, Time Spent per Session,* and *Number of Promo Hits (Others).* Since the true causal structure is not available, we estimate it using the DirectLiNGAM algorithm applied to the observational data, which also provides edge weights representing causal strengths. We then simulate data from this estimated DAG using a structural equation model (SEM) with linear relationships and additive Gaussian noise. Fig. A2 shows the causal structure with strengths estimated by DirectLiNGAM which acts as a ground truth for us.

USER LEVEL DATA - II   The USER-LEVEL DATA II dataset is constructed from the same Google Analytics Sample corpus as USER-LEVEL DATA I, but emphasizes a distinct set of behavioral features associated with user engagement. The selected variables include: *Number of Hits, Number of Unique URLs, Time Spent, Total Active Days Last Month, Number of Sessions Last Month, Number of Hits Last Month, Number of Page Hits Last Month,* and *Number of Hits on Social Network.* These features capture both cumulative and recent patterns of user interaction, providing a rich view of user behavior. Since the true causal structure is not available, as with USER-LEVEL DATA I, we estimate the underlying causal structure using DirectLiNGAM and simulate data from the resulting DAG using a linear SEM with additive Gaussian noise. This serves as the true causal structure for subsequent evaluation and experimentation. Fig. A3 shows causal structure with strengths estimated by DirectLiNGAM which acts as a ground truth for us.

CHILD   The CHILD dataset is a widely used synthetic benchmark in causal discovery. It comprises a acyclic, and semantically meaningful graph over 20 variables: *Age, Birth Asphyxia, Disease Type, Sickness, Hypoxia in $O_2$, Grunting, Lung Parenchyma, Lower Body $O_2$, $CO_2$, Chest X-Ray, LVH, Cardiac Mixing, Birth Defect, Pulmonary Stenosis, Duct flow, Cyanosis, Heart Disorder, Temperature, Heart-Rate,* and *PV SAT.* The ground-truth causal graph is shown in Spiegelhalter et al. (1993, Figure 2). For our experiments, we aggregate data from six CSV files (`child_2000_1-6.csv`) available at `https://github.com/andrewli77/MINOBS-anc/`,

Table A1: Glossary of notations used throughout the paper, categorized by their role in datasets, causal structures, parameters, expert interactions, and other components.

| Symbol | Description | Symbol | Description | Symbol | Description |
|---|---|---|---|---|---|
| **Dataset and Observations** | | | | | |
| $\mathbb{D}$ | True data distribution | $\mathcal{D}$ | Full observed dataset | $\mathbf{D}_i$ | Observed data at batch $i$ |
| $n_i$ | Number of data points in batch $i$ | $\mathbf{V}^{\mathrm{O}}$ | Set of observed variables | $d$ | Cardinality of $\mathbf{V}^{\mathrm{O}}$ |
| $\mathbf{V}^{\mathrm{L}}$ | Unobserved (latent) variables | $k$ | Look-back batch window size | $\mathcal{B}_i$ | Background knowledge till batch $i$ |
| **Causal Graphs and Structures** | | | | | |
| $\mathcal{G}$ | True underlying causal graph | $\mathbf{V}, \mathbf{E}$ | Nodes and edges in $\mathcal{G}$ | $\mathcal{G}^{\mathrm{D}_i}$ | Causal graph after batch $i$ |
| $\mathbf{V}^{\mathrm{O}}, \mathbf{E}^{\mathrm{D}_i}$ | Nodes/edges in $\mathcal{G}^{\mathrm{D}_i}$ | $\mathcal{G}^{\mathrm{X}_i}$ | Expert-provided graph till batch i | $\mathbf{V}^{\mathrm{X}_i}, \mathbf{E}^{\mathrm{X}_i}$ | Nodes and edges in $\mathcal{G}^{\mathrm{X}_i}$ |
| $A, B$ | Arbitrary variables in graph | $L$ | Latent confounder in causal structure | | |
| **Parameters and Models** | | | | | |
| $\boldsymbol{\theta}$ | Edge weights of the causal DAG | $\boldsymbol{\theta}^{\mathrm{O}}, \boldsymbol{\theta}^{\mathrm{L}}$ | Edge weights for observed, latent variable | $\phi$ | SEM parameters $\{\boldsymbol{\theta}, \sigma^2\}$ |
| $\theta_A$ | Edge weights associated with $A$ | $\phi_A$ | Parameters for $A$ variable $(\theta_A, \sigma^2)$ | $\epsilon$ | SEM noise |
| $\sigma^2$ | SEM noise variance | $\rho(A, B)$ | Correlation value between $A, B$ | | |
| **Expert Interaction and Histograms** | | | | | |
| $\alpha$ | Expert noise level | $\mathcal{H}^{\mathrm{E}_i}$ | Edge histogram till batch $i$ | $\mathcal{H}^{\mathrm{L}_i}$ | Latent variable histogram till batch $i$ |
| $\mathcal{I}^{\mathbf{E}}, \mathcal{I}^{\mathrm{L}}$ | Instructions for edge/latent expert | $\mathcal{I}^{\mathrm{P}}, \mathcal{I}^{\rho}$ | Instructions for prior/correlation expert. | $m^{\mathbf{E}}, m^{\mathrm{L}}$ | Maximum expert calls for edge/latent |
| **Others** | | | | | |
| $\mathbb{R}$ | Set of real numbers | $\mathcal{N}$ | Gaussian distribution | $\eta$ | Learning rate |
| $\lambda$ | Regularizer parameter | $\boldsymbol{m}_p$ | Prior mean | $\boldsymbol{S}_p$ | Prior covariance |

resulting in a combined dataset of 12000 samples. We convert all the variables to a categorical format producing the full dataset $\mathbb{R}^{(12000, 20)}$.

ALARM   The ALARM dataset is a widely used synthetic benchmark in causal discovery. It comprises a acyclic, and semantically meaningful graph over 37 variables. For our experiments, we aggregate data from the CSV files (`alarm_1000_1-6.csv`) available at `https://github.com/andrewli77/MINOBS-anc/`. All the variables are converted to categorical format and the details about it can be found in Beinlich et al. (1989).

RED WINE QUALITY   The RED WINE QUALITY dataset is a real-world dataset from the UCI Machine Learning Repository, comprising 11 physicochemical attributes of red wine samples—such as *density*, *alcohol*, and *residual sugar*—along with a quality rating. We utilize this dataset for *parameter estimation* experiments with linear SEMs *i.e.* of the form $A = \theta_1 B + \epsilon_i$ where $\epsilon_i \sim \mathcal{N}(0, \sigma^2)$. As shown in Fig. A5, the variable *alcohol* acts as a confounder between *density* and *quality*, making it suitable for parameter estimation evaluation. The dataset used in our experiments is sourced from the CMU Example Causal Datasets repository: `https://github.com/cmu-phil/example-causal-datasets/blob/main/real/wine-quality/data/winequality-red.continuous.txt`. It contains a total of 1599 samples.

# B   Comparisons with Other Causal Discovery Algorithms

In this paper, we focus on a setting where both *selection bias* and *latent confounding* are present. To accommodate both aspects in this general setting, it is necessary to shift from DAGs to Partial Ancestral Graphs (PAGs), since this explicitly captures uncertainty due to latent confounders and selection bias. The Fast Causal Inference (FCI) algorithm naturally aligns with these assumptions, making it a principled choice for the proposed framework, NLPSCM.

Moreover, the primary aim of this paper is not to benchmark causal structure discovery algorithms, but rather to demonstrate how NLPSCM can refine causal structure over time by integrating noisy expert knowledge (from LMs) with observational data. Notably, NLPSCM is designed to be algorithm-agnostic, and can be used in a plug-and-play manner with any PAG-generating causal discovery method. Below, we briefly discuss why several modern algorithms are not directly applicable:

1. **Greedy Equivalence Search (GES):** GES proposed by Chickering (2002) assumes both causal sufficiency and the absence of selection bias. These assumptions do not hold in our sequential setting, where both latent confounders and selection bias may be present.

2. **DAG-NoCURL and DAG-NoTEARS:** DAG-NOCURL by Yu et al. (2021) and DAG-NoTEARS by Zheng et al. (2018) are continuous optimization-based methods that output DAGs under the

assumption of causal sufficiency, without accounting for latent confounders. Thus, they do not apply to our setting, which requires a PAG-based representation to model uncertainty.

3. **DAGMA:** Bello et al. (2022) proposed DAGMA: a score-based causal discovery method that also formulates the problem as a continuous optimization task. Like DAG-NoCURL and DAG-NoTEARS, it outputs deterministic DAGs and assumes no latent confounding, making it incompatible with the sequential setting.

4. **LiNGAM:** LiNGAM (Linear Non-Gaussian Acyclic Model, Shimizu, 2014) assumes that all relevant variables are observed, *i.e.* there are no hidden confounders. This causal sufficiency assumption renders it unsuitable for the sequential setting.

5. **Recursive Causal Discovery (RCD):** Unlike FCI, RCD proposed by Maeda & Shimizu (2020) produces a DAG rather than a PAG, focusing on efficient DAG learning. Hence, when dealing with latent confounders and selection bias requiring PAG representation, RCD can not be employed.

We note that enhanced variants of FCI such as RFCI, GFCI, and FCI+ are fully compatible with NLPSCM and can be seamlessly integrated bringing their advantage to NLPSCM as well. We keep the research to study convergence impact of different algorithms for future work.

## C   Evaluation Metrics

We describe the evaluation metrics used to assess the performance of the causal discovery method.

### C.1   Modified Structural Hamming Distance (Mod. SHD)

To evaluate the structural accuracy of methods, we use a modified version of the Structural Hamming Distance (SHD) tailored for Partial Ancestral Graphs (PAGs). Unlike standard causal graphs, PAGs include multiple edge types that reflect varying degrees of causal certainty. This necessitates a more nuanced treatment of structural differences.

For example, consider the true edge $A \to B$. If one predicted PAG contains $A \leftrightarrow B$ (a definite bidirected edge) and another contains $A \circ\!\to B$ (an edge with uncertainty at one endpoint), the former should incur a higher penalty since it reflects stronger, incorrect causal commitment. That is,

$$\text{Mod. SHD}(A \to B, A \leftrightarrow B) > \text{Mod. SHD}(A \to B, A \circ\!\to B). \tag{A1}$$

Standard SHD counts the total number of missing edges, extra edges, and orientation mismatches. Our modified SHD refines the orientation mismatch term by assessing the endpoints of each edge separately. We assign a penalty of 1.0 for definite orientation errors (*e.g.,* $\to$ vs. $\leftarrow$) and 0.5 for uncertain mismatches (*e.g.,* $\to$ vs. $\circ\!\to$). This weighting scheme better reflects the confidence associated with different edge types and penalizes definitive errors more heavily than uncertainty.

### C.2   Structural Intervention Distance (SID)

The Structural Intervention Distance (SID) measures the robustness of a learned causal graph in supporting correct interventional reasoning. Unlike purely structural metrics such as SHD, SID evaluates whether the predicted graph implies the correct set of causal effects under interventions.

Formally, SID counts the number of intervention targets for which the predicted graph induces an incorrect adjustment set relative to the true graph. An SID of zero indicates that the learned structure supports identically correct interventional distributions as the ground truth, even if some edge directions are incorrect but do not affect adjustment validity.

SID is particularly useful in settings where downstream causal queries, rather than exact graph recovery, are the primary concern. It captures whether the learned structure preserves the correct set of (in)dependencies

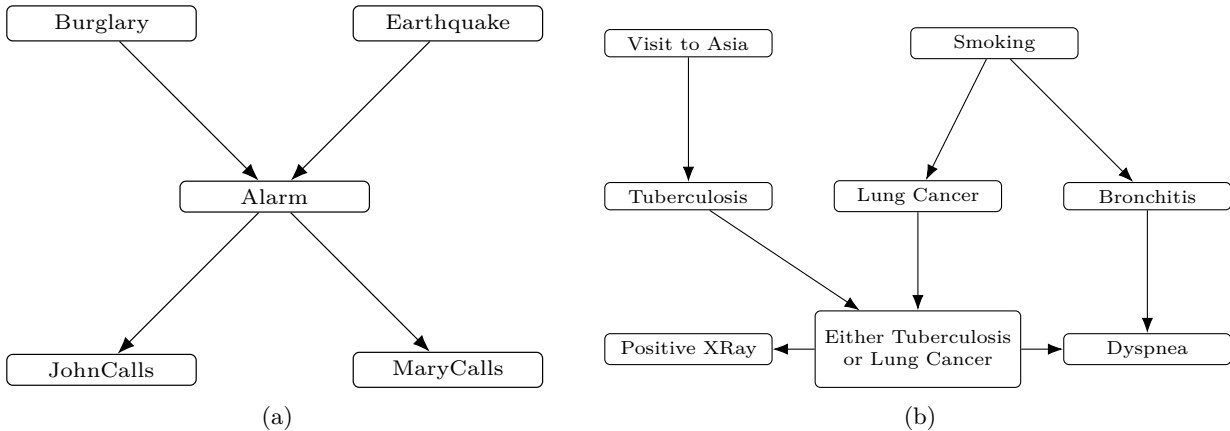

Figure A1: **Causal Structure Learning Datasets** *(a)* Ground-truth causal graph for the EARTHQUAKE dataset, where an earthquake or burglary can trigger an alarm, which in turn causes calls from John and Mary; *(b)* Ground-truth structure for the ASIA dataset, a classic dataset illustrating causal relations between variables such as smoking, lung disease, and visits to Asia.

required for estimating interventional effects, making it a practical and task-aligned evaluation metric for causal discovery.

Since SID is not directly defined for PAGs, we compute it by mapping the learned PAG into a CPDAG-compatible adjacency representation. Directed edges are preserved, while partially oriented or bidirected edges are encoded as undirected or partially specified edges. We then use the *SID_CPDAG* function from the R SID package, which computes (lower, upper) bounds on SID between a true DAG and an equivalence class. When the bounds coincide, the learned structure fully determines interventional distances. We note that this mapping is an approximation, as PAGs represent equivalence classes under latent confounding that are more general than CPDAGs, and the reported SID values should be interpreted as approximate bounds.

### C.3   Precision, Recall, F1-Score

In addition to structural metrics, we report standard classification metrics—Precision, Recall, and F1-Score—computed over predicted directed causal relationships of the form $A \to B$.

A predicted edge $A \to B$ is considered a true positive if it matches a directed edge in the ground-truth causal graph. Precision measures the fraction of predicted directed edges that are correct, while Recall quantifies the fraction of ground-truth directed edges that are successfully recovered. F1-Score is the harmonic mean of Precision and Recall, providing a balanced summary of accuracy and completeness:

$$\text{Precision} = \frac{|\text{Correctly predicted directed edges } A \to B|}{|\text{Predicted directed edges } A \to B|}, \tag{A2}$$

$$\text{Recall} = \frac{|\text{Correctly predicted directed edges } A \to B|}{|\text{Ground-truth directed edges } A \to B|}, \tag{A3}$$

$$\text{F1 Score} = 2 \cdot \frac{\text{Precision} \cdot \text{Recall}}{\text{Precision} + \text{Recall}}. \tag{A4}$$

These metrics focus exclusively on directed edges and ignore uncertain or undirected edge types. As such, they offer a more task-specific evaluation of causal directionality recovery, which is critical in downstream applications that rely on explicit causal mechanisms.

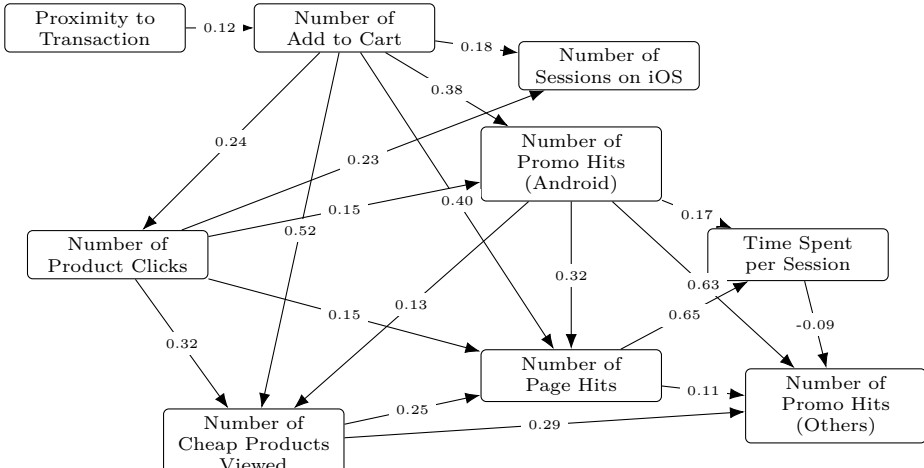

Figure A2: Ground truth causal structure for the USER LEVEL DATA - I illustrating the causal relationships between user behavioral metrics. The edge weights represent estimated causal strengths derived from observational data.

# D    Sequential Optimization as Multi-Armed Bandit

The LM interaction in NLPSCM proposed as sequential optimization (*cf.* Sec. 4.3), can presumably be interpreted as a *stochastic multi-armed bandit (MAB)* problem, providing a useful lens for understanding the design of the proposed score selection strategy (Eq. (9)).

**From Edge Querying to Bandits**    In MAB problems, a learner chooses among *K-arms*, each associated with an unknown reward distribution, and aims to maximize cumulative reward over $T$ pulls by balancing *exploration* (learning about uncertain arms) and *exploitation* (favoring known high-reward arms).

In our setting (as discussed in Eq. (6)):

- Each candidate edge $e$ is treated as an **arm**.

- Pulling arm $e$ means **querying the LM** about edge $e$'s causal type.

- **Reward** is the **information gain** from that query—specifically, the reduction in uncertainty over edge $e$'s type.

- Total LM query budget $m_L$ acts as the $T$ **horizon**.

This framing is motivated by the need to *adaptively allocate* a limited number of LM queries to edges that are either currently uncertain (exploration) or close to being included in background knowledge (exploitation). As in bandits, where arms may yield noisy feedback, LM responses are noisy, that is, stochastic and depend on context and prompt. Hence, each edge must be queried multiple times to obtain a reliable posterior.

## D.1    UCB-Inspired Selection Score

The selection score proposed in Eq. (9) resembles a classic *Upper Confidence Bound (UCB)* strategy,

$$S_i^e = w_1 E_i^e + w_2 \left( \frac{1}{TD_i^e} \right) + w_3 \sqrt{\frac{\log T_i}{T_i^e}}, \quad \text{s.t.} \quad TD_i^e = \tau_i - \max(\mathcal{H}^{\mathbf{E}_i}(e)). \tag{A5}$$

This selection score balances three factors:

- **Uncertainty (Exploration):** $E_i^e$ is the entropy of the edge's histogram—capturing how uncertain the current belief over $e$ is. High entropy indicates high uncertainty, encouraging exploration.

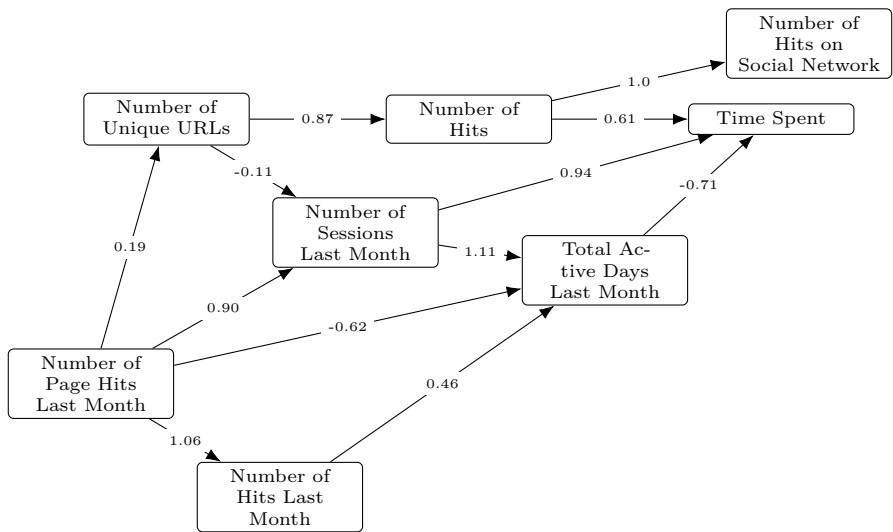

Figure A3: Ground truth causal structure for the USER LEVEL DATA - II illustrating the causal relationships between user activity metrics such as hits, time spent, session history, and social network interactions. The edge weights represent estimated causal strengths derived from observational data.

- **Threshold Proximity (Exploitation):** $1/TDe_i$ quantifies how close edge $e$ is to being included in background knowledge. A small $TDe_i$ yields a large score, encouraging exploitation of promising edges.

- **Exploration Bonus:** The term $\sqrt{\log T_i/T_i^e}$ mirrors the optimism term in UCB1. It grows slowly with total queries and shrinks with more queries to edge $e$, thus favoring edges that have not been sampled much.

Together, this scoring function implements a principled query policy that prioritizes under-explored, informative, and nearly-threshold edges. Each LM query corresponds to a pull of the edge with the highest $S_i^e$, analogous to greedy selection in UCB-based bandits.

## D.2 Regret Interpretation

Under idealized assumptions of *i.i.d.* edge-level information gains, bounded support, and stationary LM behavior, the expected cumulative *regret* after $T$ queries is:

$$\mathcal{R}(T) = \sum_{i=1}^{T}(\mu^* - \mu_{a_i}),\tag{A6}$$

where $\mu^*$ is the maximum expected gain over all edges, and $a_i$ is the edge queried at step $i$. Classical UCB algorithms ensures that

$$\mathcal{R}(T) = O\left(\sum_{e:\Delta_e>0}\frac{\log T}{\Delta_e}\right),\quad\text{where }\Delta_e = \mu^* - \mu_e.\tag{A7}$$

This sublinear growth implies that the average regret $\mathcal{R}(T)/T \to 0$ as $T \to \infty$, *i.e.* the learner asymptotically focuses on optimal arms.

Applied to NLPSCM, this suggests that if LM responses were non-stochastic, the selection score strategy concentrates LM calls on most informative edges, making increasingly efficient use of limited expert budget. However, we **caution** against this linkage to regret bounds. The above analysis rests on assumptions that do not strictly hold in the proposed NLPSCM setting due to:

- **Stochasticity:** The LM's behavior is batch-context dependent and evolves as context accumulates. Reward distributions (information gains) are not fixed.

- **Dependent arms:** Updates to the posterior of one edge can influence others due to graph constraints, violating arm independence.

- **Implicit priors:** LM outputs are influenced by prompts, prior batches, and temperature, introducing structured, non-*i.i.d.* noise.

These violations of MAB assumptions mean that classical regret bounds cannot be directly applied. Still, the MAB abstraction provides valuable intuition for designing and analyzing selection policies under uncertainty.

### D.3 Future Work

A full theoretical analysis of regret in this setting would require modeling *non-stationary, dependent* reward structures. Promising directions include *contextual and Bayesian bandits* that incorporate evolving priors, *combinatorial bandits* for structured edge dependencies, and *information-theoretic regret bounds* based on entropy reduction. We leave these extensions for future work.

## E Parameter Estimation

As outlined in Sec. 5, we propose a Bayesian parameter estimation framework designed to estimate structural parameters in the presence of latent confounders. This approach incorporates language model (LM)-suggested prior knowledge into an Expectation-Maximization (EM) algorithm. The EM-step are defined in Eq. (10) and Eq. (11).

In the experiment on the RED WINE QUALITY dataset (*cf.* Sec. 6.3), we evaluate the impact of incorporating LM-suggested priors over the latent confounder. Fig. 5 illustrates how different priors affects estimation performance, demonstrating the robustness of the proposed Bayesian method.

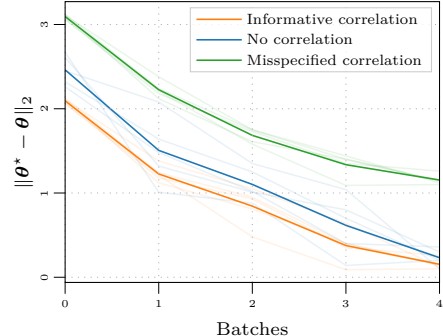

Figure A4: **Parameter Estimation:** Convergence of parameters with LM-suggested correlation as more batches are processed.

Additionally, we analyze the effect of LM-suggested correlation values $\rho(A, B)$ and investigate the robustness of the proposed parameter estimation algorithm. From the observational data we know that the confounder *alcohol* has a correlation of $\rho(\text{alcohol}, \text{density}) = -0.50$ and $\rho(\text{alcohol}, \text{quality}) = 0.48$. We set the correlation regularizer as $\lambda = 5$.

For parameter estimation algorithm, we use stochastic gradient descent with 0.001 learning rate and run the algorithm for maximum for maximum of 20 E-steps and 50 M-steps with an early stop criteria based on the expected log-likelihood value.

## F Experiment Details

This section provides detailed descriptions of the experimental setup, including sequential data setup and batching strategies, LM prompts used in the experiments, details about the FCI variants, LLM-first, ILS-CSL, and NLPSCM framework.

### F.1 Sequential Data Setup

We describe the setup for each dataset category based on the availability of ground-truth causal structure and the strategy used for batch construction.

EARTHQUAKE, ASIA These datasets provide both the true causal structure and observational data. We use the ground-truth graphs shown in Fig. A1a and Fig. A1b as the true causal structures. The observational

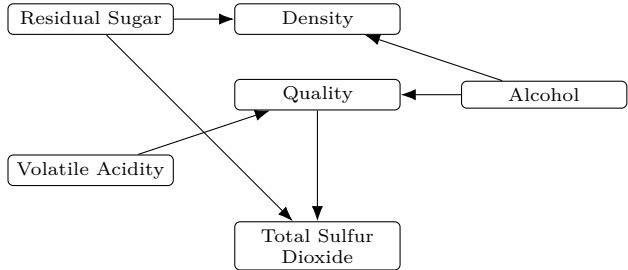

Figure A5: Ground truth causal structure for the RED WINE QUALITY dataset illustrating the directional dependencies among key physicochemical attributes affecting wine quality. The graph highlights how alcohol, volatile acidity, total sulfur dioxide, and density (influenced by residual sugar) contribute directly or indirectly to the perceived quality of red wine.

data are randomly partitioned into 6 batches of 250 samples each, without replacement. This results in data of shape $(6, 250, 5)$ for EARTHQUAKE and $(6, 250, 8)$ for ASIA. Since all variables are binary (Yes/No), we preprocess the data by mapping `Yes` to 1 and `No` to 0.

USER-LEVEL DATA I, USER-LEVEL DATA II For these datasets, ground-truth causal structures are unavailable. We apply DirectLiNGAM to the full observational dataset to estimate a causal graph, which we then treat as the true causal structure for generating synthetic data. The resulting graphs are visualized in Fig. A2 and Fig. A3. Using the estimated structure, we simulate data via a linear SEM with additive Gaussian noise, sampled from $\mathcal{N}(0, 0.05)$. We construct 7 sequential batches with varying sizes. For USER LEVEL DATA I, the batchsize is $[3000, 1000, 2000, 4000, 3000, 2000, 5000]$ while for USER LEVEL DATA II we set the batch size as $[2000, 1000, 2000, 1000, 1000, 3000, 1000]$. To simulate realistic distributions, in USER-LEVEL DATA I, the parent node *Proximity To Transaction* is sampled from $\mathcal{N}(10, 1)$ while in USER-LEVEL DATA II, the parent node *Number Of Page Hits Last Month* is sampled from $\mathcal{N}(27, 10.5)$.

CHILD, ALARM These datasets provide both the true causal structure and observational data. We use the ground-truth graphs from Beinlich et al. (1989) and Spiegelhalter et al. (1993) respectively as the true causal structures. For CHILD dataset, the observational data are randomly partitioned into 6 batches of 2000 samples each, without replacement. This results in data of shape $(6, 2000, 20)$. For ALARM dataset, we split the observational data randomly, without replacement, into six batches with 50 data points per batch. This is done with the aim to mimic the data-scarce scenario and showcase the robustness of NLPSCM.

RED WINE QUALITY This dataset provides observational data, which we use primarily to evaluate the proposed *parameter estimation* framework. We estimate the causal structure using DirectLiNGAM on the full dataset, as shown in Fig. A5. Existence of a latent confounder, *alcohol*, between *density* and *quality* makes this dataset suitable for testing latent-aware estimation. The full dataset consists of 1599 samples across 6 variables, which we partition into 5 batches of sizes $[319, 319, 319, 319, 323]$ on which we perform parameter estimation.

## F.2 LM Prompts

In this section, we provide the prompts used in the experiments.

The pairwise prompt is:

```
System message:

You are an expert in Causal discovery and are studying {experiment_name
}. You are using your knowledge to help build a causal model that
contains  all the assumptions about {experiment_name}, where a causal
model is a conceptual model that describes the causal mechanisms of a
system. You will do this by by answering questions about cause and
effect and using your domain knowledge as an expert in Causal discovery
```

```
       . We are considering the following variables: {variables}. The
       description of the variables is as follows:{variable_description}.

       User message:

       From your perspective as an expert in Causal discovery, which of the
       following is  most likely true?
       (A) {var1} affects/causes {var2}; {var1} has a high likelihood of
       directly influencing {var2};
       (B) {var2} affects/causes {var1};  {var2} has a high likelihood of
       directly influencing {var1};
       (C) Neither A nor B; There is no causal relationship between {var1} and
        {var2}.
       (D) Do not know about the causal relationship between {var1} and {var2
       }.
       Select the answer. Think step by step and provide your thoughts with
       the "thoughts" key and the answer with the "answer" key.
       Return a JSON with the following format:
       {
               "answer": "A/B/C/D",
               "thoughts": "step-by-step thought"
       }
       NOTE: Only return the JSON and nothing else.
```

The triplet prompt is:

```
       System message:

       You are an expert in Causal discovery and are studying {experiment_name
       }. You are using your knowledge to help build a causal model that
       contains  all the assumptions about {experiment_name}, where a causal
       model is a conceptual model that describes the causal mechanisms of a
       system. We are considering the following variables: {variables}. The
       description of the variables is as follows:{variable_description}.

       User message:

       As an expert in Causal discovery, consider the following variables and
       output a causal DAG, {var1}, {var2}, {var3}. Only consider direct
       causal effects. If a variable has no causal relationship with any other
       , include it as an isolated node.
       For example, if Z is independent, and X causes/affects Y, the output
       DAG should be: [["X", "Y"], ["Z"]]. Think step by step and provide your
        thoughts with the "thoughts" key.
       Return a JSON in the following format:
       {
               "dag": [["source_node", "target_node"], ..., ["isloated_node
               "]],
               "thoughts": "step-by-step thought"
       }
       NOTE: Only return the JSON object and nothing else.
```

The PAG-Pairwise prompt is:

```
       System message:

       You are an expert in Causal discovery and are studying {experiment_name
       }. You are using your knowledge to help build a causal model that
```

```
contains  all the assumptions about {experiment_name}, where a causal
model is a conceptual model that describes the causal mechanisms of a
system. You will do this by by answering questions about cause and
effect and using your domain knowledge as an expert in Causal discovery
. We are considering the following variables: {variables}. The
description of the variables is as follows:{variable_description}.

User message:

From your perspective as an expert in Causal discovery, which of the
following is  most likely true?
A: {var1} affects/causes {var2}; {var1} has a high likelihood of
directly influencing {var2};
B: {var2} affects/causes {var1};  {var2} has a high likelihood of
directly influencing {var1};
C: Neither A nor B; There is no causal relationship between {var1} and
{var2}.
D: Do not know about the causal relationship between {var1} and {var2}.
E: There is a possible latent confounder between {var1} and {var2} i.e.
 {var1} <-> {var2}
F: Can not be sure about the causal relationship however {var1} is not
an ancestor of {var2} i.e. {var1} o-> {var2}
G: Can not be sure about the causal relationship i.e. {var1} o-o {var2}
Select the answer. Think step by step and provide your thoughts with
the "thoughts" key and the answer with the "causal_option" key.
Return a JSON with the following format:
{{
            "causal_option": "A/B/C/D/E/F/G",
            "thoughts": step-by-step thought
}}
NOTE: Only return the JSON and nothing else.
```

The *LM-expert* prompt to get prior over the latent confounder

```
System message:

You are an expert in Causal discovery and are studying {experiment_name
}. You are using your knowledge to help information about the latent
confounder variable in the causal structure. You will do this by giving
 a marginal Gaussian distribution over the detected confounder variable
 along with its correlation with the connected variables. We are
considering the following variables: {variables}. The description of
the variables is as follows:{variable_description}.

User message :

We know that there is a confounder {confounder} between {variable_1}
and {variable_2}. Provide an informative marginal Gaussian on the {
confounder} between {variable_1} and {variable_2} using historical data
 and world knowledge.
Think step-by-step.
Return a JSON with the following format.
{{
            "mean": "numerical mean value of prior N({latent})",
            "variance": "numerical variance value of prior N({
            latent})",
            "correlation": "dictionary of correlation numerical
            values",
}}
```

```
                  NOTE: Only return the JSON and nothing else.
```

The *LM-expert* prompt:

```
        System message:

        You are an expert in Causal discovery and are studying {experiment_name
        }. You are using your knowledge to help build a causal model that
        contains  all the assumptions about {experiment_name}, where a causal
        model is a conceptual model that describes the causal mechanisms of a
        system. You will do this by by answering questions about cause and
        effect and using your domain knowledge as an expert in Causal discovery
        . We are considering the following variables: {variables}. The
        description of the variables is as follows:{variable_description}.

        User message:

        You are asked to determine the causal relationship between {A} and {B}.
         Only consider direct relationships and not indirect ones.The options
        available are limited to FCI output i.e. PAG edges (Use 0,2,3,4,6
        options carefully and think more about 1,2,5 as we need more of these
        options):
        0: There is no causal relationship between {A} and {B}.
        1: Changing the state of node {A} causally affects a change in another
        node {B} i.e. A->B
        2: There is a possible latent confounder between {A} and {B}
        3: Can not be sure about the causal relationship however {A} is not an
        ancestor of {B} i.e. {A} o-> {B}
        4. Can not be sure about the causal relationship, i.e., {A} o-o {B} or
        {B} o-o {A}
        5. Changing the state of node which says {B} causally affects a change
        in another node which says {A}, i.e. B->A
        6. Can not be sure about the causal relationship however {B} is not an
        ancestor of {A}, {B} o-> {A}

        Response format:
        {
                "option": option_tag,
                "thoughts": "step-by-step thought"
        }
        We know the following causal relationships: {known_relationship}
        Be extra thoughtful and careful about the relationships you are
        describing by considering both ({A}, {B}) and ({B}, {A}) before
        answering.
        NOTE: Only return the JSON object and nothing else.
```

The *LM-confounder* prompt:

```
        System message:

        Consider you are a causal discovery expert. We are creating a causal
        structure of {experiment_name} by refining the PAG obtained from fast
        causal inference (FCI) algorithm, where we are observing the following
        variables: {variables}. The description of the variables is as follows
        :{variable_description}.
        Your goal is to output a possible confounder variable name with MAXIMUM
         3 words. Be specific when providing the confounder variable and DO NOT
         provide a generic one like user behavior, etc.  In rare situations,
```

```
                    when you are not sure about the confounder, return 'undefined' as the
                    latent confounder name. Also, confounder should be outside the observed
                     variables.
                    NOTE: Be precise, and ONLY return confounder variable names with no
                    explanation.

                    User message:

                    We know that there is a latent confounder between {latent_0} and {
                    latent_1}. Can you identify the latent confounder variable name?
```

### F.3  Fast Causal Inference (FCI) Variants

In the interested setting where data arrive sequentially in batches, running standard causal discovery algorithms like FCI on the entire dataset is infeasible. To address this, we adapt the Fast Causal Inference (FCI) algorithm to sequential processing by designing multiple baselines that reflect different trade-offs. Each variant operates under a restricted lookback window size, and adapts the FCI procedure accordingly.

For all FCI-based methods, we set the significance level to $\alpha = 0.3$ for the USER LEVEL DATA II dataset, and $\alpha = 0.1$ for all other datasets. We use the *chisq* test for conditional independence on the ASIA and EARTHQUAKE datasets (due to their binary variables), and the *fisherz* test for all remaining experiments involving continuous data.

**FCI-Cumulative**  This variant applies the FCI algorithm to the cumulative data up to batch $i$, *i.e.* $\mathcal{G}^{D_i} = \text{FCI}(\mathbf{D}_{1:i})$. It serves as an upper bound on performance, assuming full access to all prior data.

**FCI-Vanilla**  This variant applies FCI independently to each incoming batch, $\mathcal{G}^{D_i} = \text{FCI}(\mathbf{D}i)$. Here, the algorithm forgets all previous knowledge, mimicking a naive local learner. This variant tests whether single-batch inference is sufficient and highlights the limitations of ignoring *data bias*.

**FCI-Iterative**  This variant introduces background knowledge by passing the output of the previous batch's FCI run as input to the next, $\mathcal{G}^{D_i} = \text{FCI}(\mathbf{D}i, \mathcal{B}_i = \mathcal{G}^{D_{(i-1)}})$. Here, the background knowledge includes previously inferred causal structure. This allows FCI to refine its structure incrementally.

**FCI-Heuristics**  In this variant, an edge from previous iterations is included in the background knowledge only if a heuristic threshold is met. Specifically, an edge is included in the background knowledge if it has appeared in the FCI outputs of at least $h$ of all the past batches (we set $h=2$ in our experiments). This strategy balances between overfitting to noise (as in FCI-Iterative) and excessive forgetting (as in FCI-Vanilla).

### F.4  LLM-first

In the **LLM-first** method, we reverse the standard causal discovery pipeline by first using a language model (LM) to propose an initial causal graph, which is then incrementally refined using sequential batches of observational data. To obtain the initial causal structure, we use a pairwise prompting strategy where the LM is queried on each variable pair. This results in a causal structure $\mathcal{G}^X$, where each edge $A \rightarrow B$ indicates a predicted causal relation. While this initial graph often captures plausible structural patterns, it suffers from overly-optimistic behavior (*cf.* Table 1) and lacks data grounding. We then sequentially refine $\mathcal{G}^X$ using batches of observed data. At each step $i$, we update the causal graph $\mathcal{G}^{X_{(i-1)}}$ using the FCI produced causal structure conditioned on the batch $i$. This adds, removes, or reorients edges based on statistical tests applied to the current data batch $\mathcal{D}_i$, thus improving the reliability of the structure over time.

### F.5  Iterative LLM-Supervised Causal Structure Learning (ILS-CSL)

Ban et al. (2023a) proposed Iterative LLM-Supervised Causal Structure Learning (ILS-CSL) algorithm that integrates natural language causal knowledge into the structure learning process by iteratively refining a causal graph using response of a large language model (LLM). Originally designed as a score-based method, ILS-CSL supervises structure learning by posing pairwise causal queries to the LLM and using its responses as

Table A2: **Causal Discovery, Impact of LM:** We experiment with GPT-4o and GPT-5 on the Earthquake and Asia dataset and report *Modified SHD* ↓ with temperature 1. We report mean and standard deviation over 5 showcasing the LM-agnostic nature of NLPSCM and superior performance across models. The inference cost of recent models, GPT-4o and GPT-5, are quite a bit more than GPT-3.5$_{\text{turbo}}$, constraining use of them for all large set of experiments.

| Dataset | Method | Mod. SHD (GPT-3.5$_{\text{turbo}}$) | Mod. SHD (GPT-4o) | Mod. SHD (GPT-5) |
|---|---|---|---|---|
| Earthquake | LLM-First | $6.00 \pm 0.82$ | $5.80 \pm 0.75$ | $5.50 \pm 0.50$ |
| | ILS-CSL | $2.38 \pm 0.96$ | $1.50 \pm 1.12$ | $1.40 \pm 1.02$ |
| | NLPSCM | $1.00 \pm 0.63$ | $0.90 \pm 0.51$ | $0.80 \pm 0.74$ |
| Asia | LLM-First | $7.33 \pm 0.94$ | $7.90 \pm 0.40$ | $7.00 \pm 1.00$ |
| | ILS-CSL | $6.50 \pm 0.50$ | $5.00 \pm 1.00$ | $5.00 \pm 0.89$ |
| | NLPSCM | $4.60 \pm 1.02$ | $3.60 \pm 1.01$ | $3.20 \pm 0.98$ |

soft constraints during graph optimization. For consistency and a fair comparison with our method, we adapt ILS-CSL to work with the FCI framework. Specifically, we treat the causal constraints inferred from the LLM as background knowledge and inject them into the FCI algorithm. This hybrid adaptation retains the core iterative supervision strategy of ILS-CSL while operating within a constraint-based causal discovery framework.

### F.6 NLPSCM

NLPSCM relies on a set of hyperparameters that guide expert interaction and edge selection throughout sequential batches. The selection score used for deciding which edge to query is computed using Eq. (9) with weights $w_1$, $w_2$, and $w_3$. Additionally, for background threshold Eq. (5), $\alpha$ and a maximum expert budget $m^{\mathbf{E}}$ per batch are used to regulate expert calls.

For the Asia, Earthquake, and User-Level Data I datasets, for selection score we set $(w_1, w_2, w_3) = (0.1, 0.6, 0.3)$, with and background knowledge threshold with $\alpha = 0.3$ and maximum expert call budgets $m^{\mathbf{E}} = 20$, $m^{\mathbf{E}} = 50$, and $m^{\mathbf{E}} = 70$ respectively. A minimum threshold of 10 is also imposed to ensure reliability. For the User-Level Data II dataset, we use $(w_1, w_2, w_3) = (0.3, 0.4, 0.3)$, with $\alpha = 0.5$, maximum expert call budget $m^{\mathbf{E}} = 20$, and a minimum threshold of 5. Table A3 showcases the NLPSCM—structure learning algorithm across LM temperatures.

**GPT-4o and GPT-5** In Table 1, we illustrate the issue of LLM over-optimism using both GPT-3.5$_{\text{turbo}}$ and GPT-4o. We find that GPT-4o does not always outperform GPT-3.5$_{\text{turbo}}$. We further demonstrate how an LM can be viewed as a noisy expert; yet can be integrated into a Bayesian causal structure discovery framework. The inference cost of recent models, GPT-4o and GPT-5, are quite a bit more than GPT-3.5$_{\text{turbo}}$, constraining use of them for all large set of experiments. That said, NLPSCM is fully model-agnostic, and stronger LMs are expected to further improve NLPSCM's performance by providing more accurate priors. As an evidence, we experiment with GPT-4o and GPT-5 model on Earthquake and Asia dataset and showcase the superior performance (Modified SHD) in Table A2.

## G Author Contributions

The core research idea was conceived by Prakhar Verma and modified in discussion with other authors during an internship with Adobe Research. Prakhar Verma was responsible for modeling the scoring rule, parameter estimation, implementing the framework and conducting the experiments. Other authors pitched in with suggestions, some of which were incorporated in the final version. The first draft was written by Prakhar Verma and Atanu R. Sinha. All authors contributed to finalizing the manuscript.

Table A3: **Causal Discovery, Impact of LM Temperature:** Table 4 shows evaluation metrics only for *temperature=1*; here we show for other two *temperatures={0.0,0.5}*. The conclusions are remarkably similar and favor NLPSCM. We show results for all six datasets using the two paradigms: *Only-Data* and *Data-LM*. We evaluate with the following metrics: *Modified SHD, SID, Precision, Recall, F1*. All the methods use GPT-3.5$_{turbo}$ as an LM-expert and we report mean and standard deviation over 5 runs.

| Dataset | Approach | Method | Mod. SHD ↓ | SID ↓ | Precision ↑ | Recall ↑ | F1 Score ↑ |
|---|---|---|---|---|---|---|---|
| EARTHQUAKE (d = 5) | Only-Data | FCI-Cumulative | $2.00_{\pm 0.00}$ | $(0.00, 5.00)_{\pm (0.00, 0.00)}$ | $1.00_{\pm 0.00}$ | $0.50_{\pm 0.00}$ | $0.67_{\pm 0.00}$ |
| | | FCI-Vanilla | $3.60_{\pm 0.80}$ | $(8.20, 9.20)_{\pm (3.60, 1.60)}$ | $0.20_{\pm 0.40}$ | $0.05_{\pm 0.10}$ | $0.08_{\pm 0.16}$ |
| | | FCI-Iterative | $5.00_{\pm 1.67}$ | $(12.20, 12.20)_{\pm (4.66, 4.66)}$ | $0.30_{\pm 0.27}$ | $0.20_{\pm 0.19}$ | $0.24_{\pm 0.22}$ |
| | | FCI-Heuristics | $3.60_{\pm 0.80}$ | $(8.20, 9.20)_{\pm (3.60, 1.60)}$ | $0.20_{\pm 0.40}$ | $0.05_{\pm 0.10}$ | $0.08_{\pm 0.16}$ |
| | Data-LM$_{(t=0)}$ | LLM-first | $6.00_{\pm 0.82}$ | $(15.00, 15.00)_{\pm (0.82, 0.82)}$ | $0.11_{\pm 0.16}$ | $0.08_{\pm 0.12}$ | $0.09_{\pm 0.13}$ |
| | | ILS-CSL | $2.00_{\pm 0.89}$ | $(5.00, 5.00)_{\pm (0.89, 0.89)}$ | $0.93_{\pm 0.13}$ | $0.55_{\pm 0.19}$ | $0.67_{\pm 0.17}$ |
| | | NLPSCM | $1.00_{\pm 0.63}$ | $(2.20, 2.20)_{\pm (1.60, 1.60)}$ | $1.00_{\pm 0.00}$ | $0.75_{\pm 0.16}$ | $0.85_{\pm 0.11}$ |
| | Data-LM$_{(t=0.5)}$ | LLM-first | $6.00_{\pm 0.82}$ | $(15.00, 15.00)_{\pm (0.82, 0.82)}$ | $0.11_{\pm 0.16}$ | $0.08_{\pm 0.12}$ | $0.09_{\pm 0.13}$ |
| | | ILS-CSL | $2.00_{\pm 0.89}$ | $(5.00, 5.00)_{\pm (2.61, 2.61)}$ | $0.93_{\pm 0.13}$ | $0.55_{\pm 0.19}$ | $0.67_{\pm 0.17}$ |
| | | NLPSCM | $1.00_{\pm 0.63}$ | $(2.20, 2.20)_{\pm (1.60, 1.60)}$ | $1.00_{\pm 0.00}$ | $0.75_{\pm 0.16}$ | $0.85_{\pm 0.11}$ |
| ASIA (d = 8) | Only-Data | FCI-Cumulative | $7.00_{\pm 0.00}$ | $(23.00, 49.00)_{\pm (0.00, 0.00)}$ | $0.00_{\pm 0.00}$ | $0.00_{\pm 0.00}$ | $0.00_{\pm 0.00}$ |
| | | FCI-Vanilla | $7.80_{\pm 0.75}$ | $(30.00, 35.00)_{\pm (5.90, 2.45)}$ | $0.00_{\pm 0.00}$ | $0.00_{\pm 0.00}$ | $0.00_{\pm 0.00}$ |
| | | FCI-Iterative | $8.00_{\pm 1.26}$ | $(33.00, 33.00)_{\pm (7.46, 7.46)}$ | $0.45_{\pm 0.24}$ | $0.23_{\pm 0.15}$ | $0.29_{\pm 0.18}$ |
| | | FCI-Heuristics | $7.80_{\pm 0.75}$ | $(30.00, 35.00)_{\pm (5.90, 2.45)}$ | $0.00_{\pm 0.00}$ | $0.00_{\pm 0.00}$ | $0.00_{\pm 0.00}$ |
| | Data-LM$_{(t=0)}$ | LLM-first | $7.33_{\pm 0.94}$ | $(27.67, 27.67)_{\pm (2.49, 2.49)}$ | $0.58_{\pm 0.12}$ | $0.29_{\pm 0.06}$ | $0.39_{\pm 0.08}$ |
| | | ILS-CSL | $5.20_{\pm 0.75}$ | $(21.80, 21.80)_{\pm (1.94, 1.94)}$ | $0.90_{\pm 0.10}$ | $0.38_{\pm 0.08}$ | $0.53_{\pm 0.09}$ |
| | | NLPSCM | $4.60_{\pm 1.02}$ | $(13.60, 13.60)_{\pm (3.83, 3.83)}$ | $0.80_{\pm 0.12}$ | $0.60_{\pm 0.12}$ | $0.67_{\pm 0.08}$ |
| | Data-LM$_{(t=0.5)}$ | LLM-first | $7.33_{\pm 0.94}$ | $(27.67, 27.67)_{\pm (2.49, 2.49)}$ | $0.58_{\pm 0.12}$ | $0.29_{\pm 0.06}$ | $0.39_{\pm 0.08}$ |
| | | ILS-CSL | $5.60_{\pm 1.36}$ | $(22.80, 22.80)_{\pm (3.19, 3.19)}$ | $0.85_{\pm 0.15}$ | $0.35_{\pm 0.09}$ | $0.49_{\pm 0.12}$ |
| | | NLPSCM | $4.60_{\pm 1.02}$ | $(13.60, 13.60)_{\pm (3.83, 3.83)}$ | $0.80_{\pm 0.12}$ | $0.60_{\pm 0.12}$ | $0.67_{\pm 0.08}$ |
| USER LEVEL DATA-I (d = 9) | Only-Data | FCI-Cumulative | $15.00_{\pm 2.77}$ | $(47.80, 47.80)_{\pm (4.62, 4.62)}$ | $0.72_{\pm 0.18}$ | $0.37_{\pm 0.09}$ | $0.47_{\pm 0.09}$ |
| | | FCI-Vanilla | $21.30_{\pm 3.50}$ | $(61.60, 61.60)_{\pm (5.54, 5.54)}$ | $0.41_{\pm 0.18}$ | $0.22_{\pm 0.10}$ | $0.29_{\pm 0.13}$ |
| | | FCI-Iterative | $5.60_{\pm 1.20}$ | $(23.40, 23.40)_{\pm (7.94, 7.94)}$ | $0.93_{\pm 0.04}$ | $0.77_{\pm 0.02}$ | $0.84_{\pm 0.03}$ |
| | | FCI-Heuristics | $21.30_{\pm 3.50}$ | $(61.60, 61.60)_{\pm (5.54, 5.54)}$ | $0.41_{\pm 0.18}$ | $0.22_{\pm 0.10}$ | $0.29_{\pm 0.13}$ |
| | Data-LM$_{(t=0)}$ | LLM-first | $8.40_{\pm 1.20}$ | $(35.20, 35.20)_{\pm (2.92, 2.92)}$ | $0.83_{\pm 0.05}$ | $0.69_{\pm 0.02}$ | $0.76_{\pm 0.03}$ |
| | | ILS-CSL | $8.50_{\pm 2.60}$ | $(30.25, 30.25)_{\pm (9.09, 9.09)}$ | $0.83_{\pm 0.08}$ | $0.69_{\pm 0.08}$ | $0.75_{\pm 0.07}$ |
| | | NLPSCM | $5.40_{\pm 0.49}$ | $(13.20, 13.20)_{\pm (2.40, 2.40)}$ | $0.88_{\pm 0.02}$ | $0.83_{\pm 0.02}$ | $0.85_{\pm 0.01}$ |
| | Data-LM$_{(t=0.5)}$ | LLM-first | $9.60_{\pm 1.69}$ | $(38.70, 38.70)_{\pm (4.94, 4.94)}$ | $0.80_{\pm 0.06}$ | $0.67_{\pm 0.03}$ | $0.73_{\pm 0.05}$ |
| | | ILS-CSL | $7.67_{\pm 1.70}$ | $(26.33, 26.33)_{\pm (3.68, 3.68)}$ | $0.84_{\pm 0.05}$ | $0.74_{\pm 0.04}$ | $0.79_{\pm 0.05}$ |
| | | NLPSCM | $4.60_{\pm 0.49}$ | $(12.00, 12.00)_{\pm (0.00, 0.00)}$ | $0.91_{\pm 0.03}$ | $0.84_{\pm 0.00}$ | $0.87_{\pm 0.01}$ |
| USER LEVEL DATA-II (d = 8) | Only-Data | FCI-Cumulative | $19.80_{\pm 2.04}$ | $(40.00, 40.00)_{\pm (3.03, 3.03)}$ | $0.15_{\pm 0.07}$ | $0.10_{\pm 0.04}$ | $0.12_{\pm 0.05}$ |
| | | FCI-Vanilla | $17.40_{\pm 1.83}$ | $(36.80, 39.40)_{\pm (2.04, 3.38)}$ | $0.07_{\pm 0.13}$ | $0.02_{\pm 0.03}$ | $0.03_{\pm 0.05}$ |
| | | FCI-Iterative | $20.60_{\pm 1.77}$ | $(42.80, 43.40)_{\pm (4.07, 4.22)}$ | $0.06_{\pm 0.08}$ | $0.05_{\pm 0.07}$ | $0.05_{\pm 0.07}$ |
| | | FCI-Heuristics | $17.40_{\pm 1.83}$ | $(36.80, 39.40)_{\pm (2.04, 3.38)}$ | $0.07_{\pm 0.13}$ | $0.02_{\pm 0.03}$ | $0.03_{\pm 0.05}$ |
| | Data-LM$_{(t=0)}$ | LLM-first | $18.33_{\pm 0.94}$ | $(42.67, 42.67)_{\pm (0.47, 0.47)}$ | $0.21_{\pm 0.04}$ | $0.19_{\pm 0.04}$ | $0.20_{\pm 0.04}$ |
| | | ILS-CSL | $19.00_{\pm 1.22}$ | $(46.75, 46.75)_{\pm (4.87, 4.87)}$ | $0.14_{\pm 0.06}$ | $0.13_{\pm 0.07}$ | $0.13_{\pm 0.07}$ |
| | | NLPSCM | $18.67_{\pm 1.70}$ | $(41.00, 41.00)_{\pm (1.63, 1.63)}$ | $0.22_{\pm 0.02}$ | $0.22_{\pm 0.04}$ | $0.22_{\pm 0.02}$ |
| | Data-LM$_{(t=0.5)}$ | LLM-first | $18.33_{\pm 0.94}$ | $(42.67, 42.67)_{\pm (0.47, 0.47)}$ | $0.21_{\pm 0.04}$ | $0.19_{\pm 0.04}$ | $0.20_{\pm 0.04}$ |
| | | ILS-CSL | $17.60_{\pm 2.33}$ | $(41.20, 41.20)_{\pm (4.12, 4.12)}$ | $0.21_{\pm 0.11}$ | $0.18_{\pm 0.12}$ | $0.19_{\pm 0.12}$ |
| | | NLPSCM | $18.67_{\pm 2.05}$ | $(41.00, 41.00)_{\pm (1.63, 1.63)}$ | $0.23_{\pm 0.02}$ | $0.22_{\pm 0.04}$ | $0.22_{\pm 0.02}$ |
| CHILD (d = 19) | Only-Data | FCI-Cumulative | $27.50_{\pm 0.00}$ | $(111.00, 131.00)_{\pm (0.00, 0.00)}$ | $0.38_{\pm 0.00}$ | $0.36_{\pm 0.00}$ | $0.37_{\pm 0.00}$ |
| | | FCI-Vanilla | $28.00_{\pm 1.48}$ | $(129.20, 133.20)_{\pm (10.46, 10.76)}$ | $0.38_{\pm 0.04}$ | $0.26_{\pm 0.05}$ | $0.31_{\pm 0.04}$ |
| | | FCI-Iterative | $32.10_{\pm 1.16}$ | $(149.00, 164.40)_{\pm (7.16, 10.33)}$ | $0.27_{\pm 0.03}$ | $0.26_{\pm 0.04}$ | $0.26_{\pm 0.03}$ |
| | | FCI-Heuristics | $28.00_{\pm 1.48}$ | $(129.20, 133.20)_{\pm (10.46, 10.76)}$ | $0.38_{\pm 0.04}$ | $0.26_{\pm 0.05}$ | $0.31_{\pm 0.04}$ |
| | Data-LM$_{(t=0)}$ | LLM-first | $29.67_{\pm 0.47}$ | $(154.83, 154.83)_{\pm (9.75, 9.75)}$ | $0.33_{\pm 0.01}$ | $0.35_{\pm 0.02}$ | $0.34_{\pm 0.01}$ |
| | | ILS-CSL | $29.80_{\pm 2.48}$ | $(160.00, 160.00)_{\pm (21.25, 21.25)}$ | $0.33_{\pm 0.05}$ | $0.35_{\pm 0.04}$ | $0.34_{\pm 0.04}$ |
| | | NLPSCM | $27.80_{\pm 0.75}$ | $(114.80, 114.80)_{\pm (7.76, 7.76)}$ | $0.39_{\pm 0.01}$ | $0.45_{\pm 0.03}$ | $0.42_{\pm 0.02}$ |
| | Data-LM$_{(t=0.5)}$ | LLM-first | $30.67_{\pm 1.11}$ | $(146.83, 146.83)_{\pm (4.81, 4.81)}$ | $0.31_{\pm 0.03}$ | $0.32_{\pm 0.03}$ | $0.32_{\pm 0.03}$ |
| | | ILS-CSL | $32.00_{\pm 2.28}$ | $(152.80, 152.80)_{\pm (12.95, 12.95)}$ | $0.28_{\pm 0.05}$ | $0.29_{\pm 0.06}$ | $0.28_{\pm 0.06}$ |
| | | NLPSCM | $28.10_{\pm 0.86}$ | $(100.60, 112.00)_{\pm (9.31, 0.00)}$ | $0.40_{\pm 0.01}$ | $0.45_{\pm 0.00}$ | $0.42_{\pm 0.01}$ |
| ALARM (d = 37) | Only-Data | FCI-Cumulative | $45.00_{\pm 0.00}$ | $(626.00, 626.00)_{\pm (0.00, 0.00)}$ | $0.25_{\pm 0.00}$ | $0.02_{\pm 0.00}$ | $0.04_{\pm 0.00}$ |
| | | FCI-Vanilla | $49.50_{\pm 1.61}$ | $(617.80, 699.20)_{\pm (29.23, 68.43)}$ | $0.00_{\pm 0.00}$ | $0.00_{\pm 0.00}$ | $0.00_{\pm 0.00}$ |
| | | FCI-Iterative | $52.40_{\pm 6.21}$ | $(612.20, 636.80)_{\pm (49.87, 43.83)}$ | $0.33_{\pm 0.14}$ | $0.12_{\pm 0.05}$ | $0.17_{\pm 0.07}$ |
| | | FCI-Heuristics | $49.50_{\pm 1.61}$ | $(617.80, 699.20)_{\pm (29.23, 68.43)}$ | $0.00_{\pm 0.00}$ | $0.00_{\pm 0.00}$ | $0.00_{\pm 0.00}$ |
| | Data-LM$_{(t=0)}$ | LLM-first | $57.33_{\pm 1.89}$ | $(695.33, 695.33)_{\pm (21.91, 21.91)}$ | $0.22_{\pm 0.03}$ | $0.10_{\pm 0.01}$ | $0.13_{\pm 0.01}$ |
| | | ILS-CSL | $49.50_{\pm 1.50}$ | $(647.50, 647.50)_{\pm (18.94, 18.94)}$ | $0.39_{\pm 0.05}$ | $0.10_{\pm 0.02}$ | $0.17_{\pm 0.03}$ |
| | | NLPSCM | $51.60_{\pm 1.77}$ | $(595.00, 595.80)_{\pm (16.63, 16.94)}$ | $0.40_{\pm 0.04}$ | $0.22_{\pm 0.01}$ | $0.28_{\pm 0.01}$ |
| | Data-LM$_{(t=0.5)}$ | LLM-first | $56.67_{\pm 2.49}$ | $(684.67, 684.67)_{\pm (16.34, 16.34)}$ | $0.23_{\pm 0.05}$ | $0.10_{\pm 0.02}$ | $0.14_{\pm 0.03}$ |
| | | ILS-CSL | $49.75_{\pm 1.48}$ | $(633.25, 633.25)_{\pm (13.05, 13.05)}$ | $0.41_{\pm 0.02}$ | $0.12_{\pm 0.01}$ | $0.19_{\pm 0.01}$ |
| | | NLPSCM | $51.70_{\pm 1.29}$ | $(594.00, 595.20)_{\pm (38.75, 38.75)}$ | $0.41_{\pm 0.03}$ | $0.22_{\pm 0.02}$ | $0.29_{\pm 0.02}$ |