# OpenReview forum: "Sequential Causal Discovery with Noisy Language Model Priors"
_TMLR — Accepted by TMLR_

### Review · Reviewer_gh5h · 2026-03-02

**Summary Of Contributions:**

This paper proposed a Bayesian causal discovery method that uses LLM to query for expert knowledge sequentially on each batch of the data as the prior to learn a causal graph in the form of the partial ancestral graph. The algorithm also has an optimization scheme to select the most informative edge when querying the language model. Specifically, the work aims to address bias introduced by language models due to hallucinations and the sampling in batches. The paper provides experiments to show that the proposed algorithm outperforms prior work.

**Additional Comments:**

N/A

**Audience:**

Yes

**Audience Explanation:**

Incorporating the output of language models as a proxy expert to provide background knowledge for causal discovery is an interesting topic.

**Claims And Evidence:**

No

**Claims Explanation:**

- Some claims lack of evidence and sound arguments.
   - The paper argues that the existing methods rely solely on access to the complete observational data or treat LMs as a primary discovery mechanism. However, the proposed method also relies on existing methods to first learn a causal structure from the first batch of the data and use LM to improve upon the resulting graph. This is described by the paragraph right above equation 1. It lacks theoretical grounding on why this is a better approach. In other words, when the causal structure learned by the existing algorithms deviates greatly from the ground truth, it is not clear how the LM experts can necessarily perform better on this object versus just treating LMs as a primary discovery mechanism. This ambiguity deserves more discussion.
    - The paper argues that the proposed hybrid framework handles both data-induced and LM-induced biases. However, the paper should discuss more about why the existing algorithms cannot handle the same problem in the same manner. For example, why can’t the FCI algorithm, combined with the background knowledge given by LMs handle the same problem?
   - It is not clear to me what the prior about the background knowledge is for the proposed Bayesian discovery method to learn a PAG even though Bayesian approach can be more sample-efficient in general.
   - The paper does not show how the output of the discovery algorithm returns a PAG. The inclusion of background knowledge seems to try to resemble a MAG instead of PAG.
   - It is not clear to me how SID is computed based on the graphical object sitting somewhere between a MAG and a PAG based on the background knowledge refinement on a PAG. I don’t know of any existing identification algorithms that evaluate such a graph object.

- The experiment shows the proposed method outperforms some baselines significantly. However, the experiment does not reflect on how modeling the data-generating process as PAG is more useful than modeling it as a DAG. The experiment also comes with concerns about the ground truth of the benchmark datasets being revealed to the language models during training, as indicated by Table 5 for the Child benchmark data.

**Requested Changes:**

- Modeling the data with a different graphical object should not be viewed as a contribution, as stated in (i). The contributions should only be about how to learn this object accurately and efficiently.
- Please address the following for Table 2:
   - What does PAG-pairwise do?
   - What does DAG-pairwise (voting) do?
   - What does temp. mean?
    - Why are there upper and lower bounds for SID? Why are they meaningful if the upper bound is the same as the lower bound?
- Please describe how to set up the prior for the Bayesian causal discovery approach. More specifically, how $p(G_{i}|\mathcal{B}_{i-1})$ is derived and how is $p(\mathcal{B}_{0})$ is set up ?
- Please address how FCI algorithm in design fails to handle both data-induced bias and the LM-induced bias.
- Please address the issue why table 2 presents a comparison between DAG and PAG, but not DAG with maximal ancestral graphs (MAG).
- ‘Any categorical value for an observed variable is encoded as a numerical value.’ Please describe why this is a reasonable choice.
-  It is not clear why a PAG is modeled but not MAG while the prior work has been modeling a DAG not a CPDAG. The authors should address this issue.
- Please explain how $\alpha$ is selected for equation 5 and the intuition of the equation of $\tau_{i}^{e}$. Also, it would be great to talk about why $\sqrt{T_{i}^{e}(1- \frac{T_{i}^{e}}{T_{i}})}$ is the symmetry of $E_{i}^{e} \times T_{i}^{e}$ since they are controlled by $\alpha$.
- It would be better to introduce what $m^{E}, m^{L}$ refer to before introducing algorithm 1.
- The paper should discuss whether the LMs used in the experiments have seen the ground truth from the well-known benchmarks from their training data.
- The table 4 formatting is inconsistent. It sometimes highlights multiple algorithms and sometimes only highlight the proposed algorithm.
- It is unclear how the reward is computed given an edge in the sequential optimization step.
- The paper should include a proof to show the output the algorithm is indeed a PAG.
- The paper should explain why it is critical to estimate the edge weights and noise variance.
- The paper should explain why the DAG inferred by DirectLiNGAM can be treated as the ground truth in the experiment for the two USER-LEVEL datasets.
- The paper should add citations to each algorithm described in Section B in the Appendix.
- The paper should include a baseline that simply uses FCI combined with background knowledge from the language models.
- The paper should explain how Table 4 reflects the performances of each algorithm on each batch of the data in each of the repeated experiments. It is not clear whether the mean reported is an average over the averages evaluated based on each batch of the data.
- The experiment does not reflect on how modeling the data-generating process as PAG is more useful than modeling it as a DAG. The experiment should try to randomly hide the confounders in the ground truth to make the contributions more meaningful.
- The paper should describe more explicitly how LLM-first combines the output of FCI with the prior given by LLM. Does it remove the contradictory edges or simply replace it whenever the FCI output disagrees? It is not clear.

---

> ### Author Response · Authors · 2026-03-25
> **Author Response (1/5)**
>
> We thank the reviewer for their insightful comments on the manuscript. We are encouraged that the reviewer finds the topic of incorporating LM outputs as proxy expert knowledge for causal discovery interesting and relevant to the TMLR audience. Below, we address the concerns raised regarding the support for claims and the requested changes. We note that several questions relate to a common theme of the behavior of FCI under background knowledge and the validity of PAG as the output representation. We address these points and concerns below.
>
> **Support for claims.**
> * **Data-first vs. LM-first approach.** The key motivation for the hybrid design is that data-driven methods and LMs provide complementary sources of information. FCI extracts statistically grounded structure from observational data, but can be unreliable for a batch of data due to sample selection bias and small-sample effects. In contrast, LMs provide global, semantic priors that are independent of these effects but may be noisy or biased in another way.
> \
> The role of FCI in NLPSCM is therefore not to provide a fully accurate initial graph, but to constrain and structure the LM interaction space, allowing queries to focus on statistically plausible edges rather than exploring the full graph space. This significantly improves the informativeness of LM queries compared to treating the LM as a primary discovery mechanism.
> \
> Empirically, this design choice is directly evaluated in Table 4 and Table 5. The *LLM-first* baseline treats the LM as the primary discovery mechanism, while *FCI-Vanilla* applies FCI independently on each batch. NLPSCM consistently outperforms both across datasets, indicating that even noisy data-driven structures, when refined through sequential LM interaction, are more effective than either approach alone. Additionally, the sequential formulation provides a correcting mechanism, where background knowledge accumulated across batches progressively improves structural estimates, mitigating errors from early batches.
>
> * **FCI with LM-derived background knowledge** A key limitation of directly combining FCI with LM-derived background knowledge is that such approaches typically treat the background knowledge as fixed and reliable constraints, whereas LM outputs can be noisy and inconsistent. As a result, directly incorporating LM suggestions into FCI can propagate errors or introduce contradictions without a mechanism to resolve uncertainty or adapt over time.
> \
> NLPSCM addresses this by explicitly modeling LM outputs as noisy observations, rather than deterministic constraints. Instead of immediately enforcing LM-derived edges, NLPSCM accumulates evidence across batches via histograms and promotes information to background knowledge only when sufficient confidence is reached through a dynamic thresholding mechanism. This allows the method to mitigate both LM-induced noise and data-induced variability in early batches.
> \
> The closest existing approach to the reviewer’s suggestion is ILS-CSL (Ban et al., 2023a), which combines causal discovery with LM-derived background knowledge at each batch. However, NLPSCM differs in several key ways: *(i)* it performs uncertainty-aware aggregation of LM responses rather than direct incorporation, *(ii)* it uses a sequential query optimization strategy (Eq. 9) to focus on the most informative edges under a limited query budget, and *(iii)* it operates over PAGs, allowing uncertainty and partial orientation to be represented explicitly. Moreover, empirically, NLPSCM consistently outperforms ILS-CSL across all datasets and LMs (Tables 4, 5), and ablations in Fig. 4 show that both the uncertainty-aware thresholding and query selection contribute significantly to these gains from using NLPSCM.

---

> ### Author Response · Authors · 2026-03-25
> **Author Response (2/5)**
>
> * **Bayesian Prior over background knowledge.**  The point is well taken. In stating (Page 2, paragraph 2) that *"the causal structure discovery itself is Bayesian in that the beliefs about causal structure from data are updated iteratively by information drawn from an LM, as data arrive in batches,"* we were imprecise. We should have said that this belief updation across sequential batches is Bayesian-inspired. We agree that the causal-structure learning algorithm of the framework does not maintain a full posterior distribution over causal structures and thus is not Bayesian in the classical structure learning sense. The intended use of the Bayesian-inspired framing (Eq. 3) is that it serves as a conceptual foundation for integrating noisy LM information over batches. In particular, we treat LM outputs as noisy, prior information and accumulate evidence across batches. We maintain edge-level belief summaries (via histograms) which can be interpreted as sequential estimation of edge-type distributions under noisy observations. These updates are analogous to Bayesian evidence accumulation and provide a principled, uncertainty-aware approximation at the edge level, even though they do not maintain a posterior over the graph space. NLPSCM produces a PAG augmented with uncertainty-aware edge beliefs that is updated efficiently across batches. We also note that our histogram-based formulation is closely related to recent work on semantic entropy in language models (Section 4.2) which similarly aggregates uncertainty across samples.
> \
> We have revised the manuscript accordingly, and now describe the structure-learning algorithm as Bayesian-inspired (Page 2,Page 6 and Section 4.2) and make explicit that uncertainty is maintained at the edge level rather than as a posterior over graphs (Page 6 and Section 4.2). We have also mentioned the fully probabilistic inference over graph structures as an important direction for future work (Page 13, Limitation and Future Work).
>
> * **PAG vs MAG under background knowledge.** The FCI algorithm outputs a PAG, which represents a Markov equivalence class of MAGs. When background knowledge is incorporated, it acts as a set of constraints that restrict this equivalence class, but the output remains within the PAG space [1]. In our implementation (via causal-learn), background knowledge is passed directly into the FCI procedure as required/forbidden edges and orientations, rather than being applied post hoc. As a result, the discovery process continues to produce a PAG consistent with both the data and the imposed constraints, rather than collapsing into a single MAG.
> \
> [1] Andrews, B., Spirtes, P., & Cooper, G. F. (2020). On the Completeness of Causal Discovery in the Presence of Latent Confounding with Tiered Background Knowledge. Proceedings of the Twenty Third International Conference on Artificial Intelligence and Statistics (AISTATS), PMLR 108:4002-4011.
>
> * **SID computation.** We thank the reviewer for this important question. We agree that SID is not directly defined for PAGs, and that this was not clearly explained in the manuscript. In our implementation, we do not compute SID directly on the learned PAG. Instead, we map the PAG into a CPDAG-compatible adjacency representation and use the *SID_CPDAG* function from *R* package, which computes lower and upper bounds on SID between a true DAG and an equivalence class. Concretely, different PAG edge types (*e.g.,* partially oriented or bidirected edges) are encoded into a format compatible with CPDAG-based SID computation. This mapping preserves directed edges and represents uncertainty through undirected or partially specified edges, allowing us to evaluate SID bounds over DAGs consistent with the learned structure. We note that this mapping is an approximation, as PAGs represent equivalence classes under latent confounding, which are more general than CPDAGs. As a result, the reported SID values should be interpreted as approximate bounds reflecting the quality of the learned structure rather than exact SID defined for PAGs. We note that our other evaluation metrics, Modified SHD, Precision, Recall, and F1-score, are computed directly on the PAG structure without requiring this mapping, and NLPSCM's improvements are consistent across all metrics. We have revised the manuscript (Page 21, App. C.2.) to explicitly describe this evaluation procedure.

---

> ### Author Response · Authors · 2026-03-25
> **Author Response (3/5)**
>
> * **Empirical benefit of PAG over DAG.** The motivation for using PAGs is described in Sec. 4.1, where we argue that PAGs allow representation of latent confounding, partial orientation, and uncertainty, which are central to our setting involving biased batch-data and noisy LM-derived knowledge. The choice of PAG over DAG is not merely an empirical preference but a modeling requirement. DAGs make a restrictive assumption of causal sufficiency, which we relax as called upon by our problem setup. This is also empirically supported in Table 2, where we compare DAG-based and PAG-based prompting strategies (DAG-Pairwise *vs.* PAG-Pairwise) and observe improved SID when using PAG representations.
>
> * **LLM training data leakage.** We explicitly discuss this concern in Sec. 6.2. In particular, we note that benchmark datasets such as Child are likely to have been seen during LM training, and we introduce BFS as a baseline to examine reliance on LM knowledge. We further contrast this with User Level Data-I, which is unlikely to be present in training data, and observe that NLPSCM continues to outperform BFS, with a larger margin on this dataset. These results suggest that the gains are not solely due to memorization of benchmark structures.
>
> **Requested Changes**
>
> * **PAG as representation in hybrid LM setup.** We clarify that our contribution is not merely adopting a different graphical object, but recognizing that existing LM-augmented causal discovery methods universally operate over DAGs, which cannot represent the uncertainty and latent confounding inherent in the hybrid setting. Prior work forces LM responses into DAG format, requiring post-hoc heuristics to resolve inconsistencies, a limitation documented in Table 1. The shift to PAGs enables a coherent framework where these are represented natively.
> * **Table 2 clarifications.** *DAG-Pairwise (Voting)* queries the LM for each variable pair with the standard pairwise prompt, samples multiple responses, and uses majority voting to determine each edge, producing a DAG. *PAG-Pairwise* similarly queries each variable pair but allows the LM to select from the full PAG edge type set $\mathbf{E}^{PAG}$, enabling the LM to express uncertainty and partial orientation. *Temp.* refers to the LM sampling temperature. *SID* is reported as (lower, upper) bounds over the equivalence class represented by the learned structure. When the bounds coincide, the structure fully determines interventional distances with no residual ambiguity.
> * **Setting up Prior.** As discussed in our response to the prior over background knowledge above, $p(G_i \mid B_{i-1})$ is operationalized through hard constraints derived from cumulative histograms via the dynamic threshold (Eq. 5), passed to FCI to restrict the admissible PAG space. For initialization, $B_0 = \emptyset$, *i.e.,* no background knowledge is assumed at the first batch, and FCI runs unconstrained on $D_0$. The histograms $HE_0$ are similarly initialized as empty. This allows the framework to be entirely data-driven initially and progressively incorporate LM-derived evidence as batches accumulate.
> * **FCI and dual biases.** FCI is a purely constraint-based algorithm that performs conditional independence tests on the provided data. It has no mechanism to incorporate external knowledge sources, so LM-induced bias is outside its scope entirely. For data-induced bias, FCI treats each dataset it receives at face value *i.e.,* it has no built-in mechanism to account for finite-sample unreliability or accumulate structural evidence across batches. This is reflected in Table 4, where FCI-Vanilla and FCI-Cumulative both perform poorly compared to NLPSCM. NLPSCM addresses these limitations by layering uncertainty-aware LM integration and sequential evidence accumulation on top of FCI's statistical foundation.
> * **DAG vs PAG vs MAG in Table 2.** A MAG represents a single ancestral graph, requiring all edges to be fully determined—similar to DAGs in this regard. A PAG represents the equivalence class of MAGs, preserving ambiguity through edge types such as $\cdot\rightarrow$ and $\cdot-\cdot$. In our setting of noisy LM responses and limited, potentially biased batch data, committing to a single MAG would reintroduce the same overcommitment problem that motivates our departure from DAGs (cf. Table 1).

---

> ### Author Response · Authors · 2026-03-25
> **Author Response (4/5)**
>
> * **Categorical encoding.** This is standard practice in causal discovery with conditional independence testing. FCI requires numerical input for statistical tests such as Fisher's Z. Categorical variables are encoded numerically (*e.g.,* ordinal or indicator encoding) to enable these tests. This choice is consistent with common implementations in the causal-learn library and does not affect the structural discovery procedure, as FCI operates on conditional independence relations rather than raw variable values.
> * **PAG vs MAG and the analogy to DAG vs CPDAG.** Prior LM-augmented methods model DAGs rather than CPDAGs precisely because their prompting strategies force definite edge determinations, with heuristics to resolve resulting inconsistencies. We argue this is a limitation, not a principled choice. In our setting with latent confounders, the analogous limitation would be modeling MAGs instead of PAGs, again forcing full edge resolution where the evidence does not support it. NLPSCM chooses PAGs over MAGs for the same reason one would choose CPDAGs over DAGs: to preserve the equivalence class structure and represent uncertainty faithfully. Our use of FCI is to output PAGs by design, and our framework is built to leverage this representational capacity.
> * **Equation (5).** The threshold balances two sources of uncertainty. The first term captures distributional uncertainty *i.e.,* how spread the histogram is over edge types—scaled by the number of interactions. The second term captures sampling uncertainty, that remains large when edge $e$ has been insufficiently queried relative to total queries. That is, the first term addresses what the evidence says while the second addresses how much evidence we have, with $\alpha$ controlling this trade-off. We select $\alpha$ via grid search.
> * **Notation $m^E$, $m^L$.** These are defined in Sec. 4.3 on the same page as Algorithm 1 as well as in the Table A1. We have now added a Notation text below Algorithm 1.
> * **LM training data leakage.** This is discussed in Sec. 6.2 of the manuscript. Please see our response under "LLM training data leakage" above.
> * **Table 4 formatting.** The highlighting reflects statistical significance: we highlight all methods that are not significantly different from the best-performing method under a two-tailed test, as noted in the table caption.
> * **Reward computation in sequential optimization.** The reward in Eq. (7) is operationalized through the scoring function $S_i^e$ in Eq. (9), which combines three components: the entropy $E_i^e$ of the edge's histogram (uncertainty about edge type), the inverse threshold distance $1/TD_i^e$ (proximity to being promoted into background knowledge), and an exploration term $\sqrt{\log T_i / T_i^e}$ (encouraging queries to under-explored edges). The edge with the highest score is selected for the next LM query. We have revised the manuscript to make the connection between the reward formulation (Eq. 7) and the implementation (Eq. 9) more explicit (Page 8).
> * **Output is a PAG.** FCI is proven to be sound and complete, outputting a valid PAG, including when background knowledge is incorporated (Andrews et al., 2020). Since NLPSCM uses FCI as its discovery algorithm and passes background knowledge through the standard interface, the output at each batch is guaranteed to be a valid PAG by construction. Therefore, no additional proof is required beyond the existing theoretical guarantees of FCI.
> * **Edge weight and noise variance estimation.** Causal structure discovery identifies which causal relationships exist, but for downstream tasks, such as estimating causal effects or policy evaluation, the strength of these relationships is essential. Edge weights quantify causal effect magnitudes, and noise variance characterizes unexplained variability. Without parameter estimation, the discovered structure remains qualitative and cannot be used for quantitative causal tasks.
> * **DirectLiNGAM as ground truth.** DirectLiNGAM is applied to the full dataset to infer a DAG, exploiting non-Gaussianity to identify a unique causal structure rather than an equivalence class. We then simulate sequential batch data from this inferred DAG, ensuring consistency between the ground-truth structure and the data used for evaluation. This approach follows standard practice when ground-truth causal structures are unavailable, a common phenomenon in causal discovery research.
> * **Citations in Section B.** We have added appropriate citations for each algorithm described in Sec. B in the appendix.

---

> ### Author Response · Authors · 2026-03-25
> **Author Response (5/5)**
>
> * **FCI with LM background knowledge baseline.** As discussed in our response above, the closest existing baseline to this suggestion is ILS-CSL (Ban et al., 2023a), which iteratively combines causal discovery with LM-derived background knowledge. NLPSCM consistently outperforms ILS-CSL across all datasets and LMs (Tables 4, 5).
> * **Table 4 reporting.** Table 4 reports metrics evaluated on the final batch, averaged over 5 independent runs (with standard deviation).
> * **PAG vs DAG and hidden confounders.** The first part is addressed in our response on *"Empirical benefit of PAG over DAG"* above. Regarding the suggestion to hide confounders: this is precisely what we do in the red-wine quality experiment (Sec. 6.3). The variable *alcohol_content* is treated as unobserved, NLPSCM's structure learning identifies a latent confounder between quality and density, and the LM correctly identifies alcohol_content as the top candidate (Fig. 6). The parameter estimation then incorporates this latent confounder and recovers accurate edge weights (Fig. 5). This experiment directly demonstrates the benefit of PAG-based modeling, as DAGs cannot represent latent confounding.
> * **LLM-first implementation.** In the LLM-first baseline, the LM is first queried to produce a causal structure, which is then passed as background knowledge to FCI. FCI runs with these constraints and the observational data, producing the final output.

---

### Review · Reviewer_2Teq · 2026-03-16

**Summary Of Contributions:**

The authors propose a method for language model (LM)-augmented observational causal discovery, under the following problem setting: (1) data is sampled from the marginal distribution over variables $\mathbf{V}^O \subseteq \mathbf{V}$, where the distribution of $\mathbf{V}$ comes from a linear Gaussian causal graphical model, (2) data arrives in batches, and (3) after receiving each batch of data, the user can make $m^L$ calls to an LM, which is treated as a noisy/imperfect expert. In particular, the method returns a partial ancestral graph (PAG) along with estimated parameters, and the LM provides (imperfect) query access to several graph features, including (1) the "type" of edge connecting any two variables *var1* and *var2*, where the "type" can be any of the 7 kinds of edges in a PAG, and (2) the prior for a latent confounder, as a conditional Gaussian distribution given the two variables which it is confounding.

The contributions include the introduction of this method, called *Noisy Language Prior in Sequential Causal Modeling (NLPCSM)*, along with several experiments that compare this method to "Only-Data" methods (varieties of FCI) and other "Data-LM" methods (e.g. ILS-CSL). The NLPSCM method includes several subcontributions, including the use of PAGs instead of DAGs (as already mentioned), the use of a dynamic thresholds for edge inclusion (inspired by the multi-armed bandits literature), and an expectation-maximization algorithm for parameter estimation. The experiments involve variety along several dimensions:
1. **Datasets:** The experiments include standard bnlearn datasets (EARTHQUAKE, ASIA, CHILD, and ALARM) and two real-world datasets (USER LEVEL DATA I AND II), which are used in a semisynthetic manner.
2. **Language models:** The experiments are conducted on several language models, including GPT-3.5 (turbo), LLaMa3.1 (8B-instruct), Qwen3 (4B-instruct), and GPT4.1 (nano), spanning different parameter counts, training pipelines, and open- vs. closed-source models. Further, the results are reported across a variety of temperature parameters.
3. **Evaluation metrics:** The experimental results report several structure-based evaluation metrics, including a PAG-modified Structural Hamming Distance (Mod SHD), Precision, Recall, F1 Score, and Structural Intervention Distance (SID).
Across the experiments, the authors find that their method consistency outperforms baselines on all metrics, with minor exceptions (e.g. one variant of FCI outperforms their method on the ASIA dataset in terms of Mod SHD), with proper reporting of standard deviations and significance levels over 5 runs of each method.

### Strengths
1. **Fairly extensive experiments:** The experiments are fairly extensive, including an appropriate range of datasets, language models, and appropriate evaluation metrics.
2. **Thoughtful method design:** The NLPSCM method is thoughtfully designed to handle the imperfection of language models as experts (e.g. helping to address known issues such as hallucination), the sequential nature of inference (e.g. using histograms as sufficient statistics for the expert responses) and real-world issues such as selection bias and unobserved confounding (e.g. by using PAGs as a representation, which are by construction able to represent these issues). Further, the framing of the method as a posterior inference problem allows the numerical data and the LM responses to be treated in a unified, principled fashion.
### Weaknesses
1. **Presentation issues:** The paper has some minor-to-medium presentation issues, e.g. ordering of figures and tables and a possible error in Fig. 4 (see Requested Changes #1), missing explanations for some key details (see Requested Changes #2, #3.1, #3.2), and some vague/imprecise/misleading language (see Requested Changes #4).
2. **Methodological limitations:** While thoughtfully designed, the method still has important limitations, some of which were not addressed in the paper. In particular, the method seems to assume that any unobserved confounders are only unobserved confounders for a *pair* of variables, rather than multiple variables (see Requested Changes #3.3). Other main limitations (e.g. the assumption of linear Gaussian models) are an okay starting place, but could be more explicitly mentioned in the abstract and introduction.
3. **Hyperparameter selection:** The main detail that seemed to be missing about the experiments was hyperparameter selection for the different methods. The hyperparameters for their method (NLPSCM) and for the Only-Data methods are stated in the appendix, but there were no details on how these hyperparameters were selected (it appears they were arbitrarily chosen). Meanwhile, the hyperparameters for the Data-LM methods were not explicitly mentioned even in the appendix. For fair comparison, proper hyperparameter tuning should be performed for all methods (see Requested Changes #5).

**Audience:**

Yes

**Audience Explanation:**

Yes, causal discovery broadly, and LM-augmented causal discovery in particular, is a good fit for the TMLR audience. The TMLR audience would be especially interested in the principled framing of this problem as a form of posterior inference, the treatment of LMs as imperfect experts, the details of the proposed method (e.g. the prompting strategy and the use of dynamic thresholding inspired by multi-armed bandits), and the solid experimental results.

**Broader Impact Concerns:**

The paper does not include a Broader Impact statement, but it should add one touching upon at least the following issue:
- **Downstream use of learned causal models:** If the causal models learned by this method are used for downstream decision-making, there are several standard concerns (e.g. model misspecification) that should be taken into account. Any downstream uses of the learned models should be subject to careful, responsible conduct.

**Claims And Evidence:**

Yes

**Claims Explanation:**

The main claim - that the new method outperforms prior work on varied datasets and LMs - is supported by fairly extensive experiments. However, there is an issue with this evidence, namely the lack of fair hyperparameter selection.

**Requested Changes:**

1. **Figures:** The paper has some minor issues with the figures. Please indicate how these will be addressed:
	1. *Rearrange figures/tables:* The placement of figures, tables, and algorithms was less than ideal, e.g. Fig. 4 is referenced before Table 3, but appears far after Table 3, and the references are far apart from the actual displays (e.g. Fig. 4 is referenced on page 6 but is on page 10). As much as possible, the figures should be within one page of where they are referenced, and the figures should appear in the order they are referenced.
	2. *Check Figure 4(right):* In the text (last paragraph of 6.1), it is stated that the dynamic threshold outperforms a conventional fixed threshold. However, in the figure, the blue line (fixed threshold) has lower Mod SHD than the red line (dynamic threshold). I assume this is a plotting issue, please confirm.
2. **Additional explanations:** Several details could use more explanation. Please clarify these points and discuss the corresponding proposed revisions:
	1. *PAG pairwise prompting:* I have worked with PAGs and understand the semantics of the edges, but I did not understand how these edges are output from the main paper. After reading Appendix F.2, I see how this step is done, but it is an important enough detail that it should be briefly be described in the main paper.
	2. *Tuple for SID:* In all tables, SID appears with two values $(lower, upper)$, but I did not catch anywhere where this range was defined (even in the appendix). Please describe what this range indicates.
	3. *Motivation for threshold expression:* I understand that the threshold in Equation (5) is motivated from the multi-armed bandit literature, but this perspective is only introduced afterwards (in Section 4.3). When reading from top to bottom, it feels as though these expressions come out of nowhere, with little intuition for *why* to use the threshold in such a form. It would be helpful if the MAB perspective was introduced earlier, along with the explore-exploit tradeoff and citations to related works from which these expressions are motivated, before introducing the expression themselves.
3. **Discuss limitations around latent confounders:** The main paper includes little detail about how latent confounders are introduced and dealt with, and upon looking through the appendix, I have more questions. Please clarify these points and propose corresponding changes to the paper:
	1. *Confounder name generation:* In the LM-expert prompt (to get the prior over the latent confounder) in F.2, and in the example in Table 6, it appears important that the confounder is given a semantically-meaningful name, e.g. "alcohol content". However, the PAG-Pairwise prompt does not clearly show whether this name is generated, and if so, how it is generated. Does the algorithm automatically suggest a name, or does it require a human in the loop? For the example in Section 6.3, the confounder name seems to be a manual input, is this true in general? It is okay if so, but should be explicitly mentioned in Section 6.3.
	2. *Mismatch between unconditional/conditional distribution:* The LM-expert prompt is supposed to return a conditional distribution over the latent confounder given the two observed variables, but Table 6 gives an unconditional distribution. Was this distribution derived from the conditional distribution, or is there some other error (e.g. a typo or an error in the LM's output)?
	3. *Only pairwise confounding:* In the LM-expert prompt (to get the prior over the latent confounder) in F.2, it appears that the algorithm only allows for pairwise latent confounding, but latent confounding in PAGs may reflect the marginalization of a confounder which affects several observed variables. This limitation should be explicitly stated and the authors should provide some justification.
4. **Modify imprecise/misleading language:** There were a few passing statements that were somewhat incorrect, though were not central to the paper's message. Please respond to these if you disagree, or acknowledge and propose a change:
	1. *Availability of domain experts:* In the abstract, you state "causal discovery... typically assumes access to... availability of domain experts". However, most causal discovery algorithms (e.g. PC, LiNGAM, FCI, NOTEARS) are completely data-based and do not assume any access to domain experts; though some algorithms do provide support for inputting domain expertise (e.g. "fixed edges" or "fixed gaps" in some PC/FCI implementations). As someone who has been in the field of causal discovery for a while, I would not agree with this statement; rather I might say something more mild, e.g. "causal discovery algorithms supporting input from domain experts typically assume perfect expertise, though recent methods have been proposed for imperfect experts".
	2. *Bias in batch data:* In several places (e.g. last paragraph of page 4, "the inherent bias in batch data"), it is suggested that there is some kind of confounding/selection bias, where data in different batches would be sampled from different distributions. However, in Section 3 (Problem Setup), it is assumed that each batch is sampled from the same underlying true distribution. Hence, I did not understand the intention behind the term "data-induced bias": either the Problem Setup should explicitly allow for the possibility of different distributions for each batch, or a different term should be used if you are simply referring to estimation error from finite data (which is not "bias" in the statistical sense, but closer to "variance").
	3. *Uninformative priors:* In Section 6.3, you state "when queried with obscure variables... LM often defaults the the unit Gaussian... akin to using an uninformative prior". However, in Bayesian inference, the term "uninformative prior" is typically meant to refer to a prior with very large spread, e.g. a Gaussian with very large variance. The unit Gaussian would typically *not* be referred to as an uninformative prior. This connection feels unnecessary anyways, so I would suggest removing it.
5. **Add fair hyperparameter selection:** Can you ensure that hyperparameter selection is performed fairly for all methods? In particular, some standard hyperparameter tuning (e.g. through `optuna`) should be conducted to choose the final hyperparameters for each algorithm, this tuning should be described in the main paper, and the final evaluation metrics should be computed on a separate validation set.

---

> ### Author Response · Authors · 2026-03-25
> **Author Response (1/2)**
>
> We thank the reviewer for their insightful comments on the manuscript. We are encouraged that the reviewer finds the method design thoughtful, the experiments fairly extensive, and the problem framing of interest to the TMLR audience. The weaknesses identified (presentation, methodological limitations around latent confounders, and hyperparameter selection) are directly addressed through the requested changes below.
>
> * **Figure/table placement.** We acknowledge that the current placement is suboptimal. Due to the density of tables, figures, and algorithms, achieving ideal placement within the current page limits is challenging. We will make best efforts to improve placement in the final version of the manuscript.
> * **Figure 4 (right).** The reviewer is correct: the legend colors are swapped. NLPSCM-threshold (dynamic) corresponds to the blue line, similar to the left and middle figure where blue is NLPSCM. We thank the reviewer for catching this and have updated the manuscript (legend in Figure 4(right)).
> * **PAG pairwise prompting.** We agree this is an important detail. We have added a brief description of the PAG pairwise prompting procedure in the main paper (Sec. 4.1, Page 6), and refer to Appendix F.2 for the full prompt.
> * **Tuple for SID.** The two values represent (lower, upper) bounds on SID, computed over the equivalence class represented by the learned structure. When the bounds coincide, the structure fully determines interventional distances with no residual ambiguity.
> * **Motivation for threshold expression.** We agree that the threshold expression appears without sufficient motivation in the manuscript. We have updated the paragraph before Eq. (5) explaining why the threshold balances distributional uncertainty and sampling uncertainty, with a forward reference to Sec. 4.3 where the sequential optimization framework is fully developed.
> * **Confounder name generation.** The confounder name is generated entirely by the LM, no human input is required. Fig. 6 shows the histogram of LM-predicted confounder candidates (*e.g.,* alcohol_content, grape_quality, grape_maturity), all suggested by the LM. The prompt for this step was inadvertently omitted from App. F.2. We have updated the manuscript (Page 28, Appendix).
> * **Unconditional/conditional distribution.** The reviewer is correct to flag this inconsistency. The EM algorithm (Alg. 2, Eq. 10) requires a marginal prior $p(V_L)$, not a conditional distribution. Table 6 correctly reports this marginal prior, *e.g.,* $\mathcal{N}(12.5, 2.5)$ for alcohol content. The prompt description in Appendix F.2 incorrectly refers to as a "conditional distribution," which we have corrected in the manuscript (Page 27, Appendix).
> * **Pairwise confounding.** The current implementation assumes each latent confounder affects exactly two observed variables. This is a simplifying assumption we adopt to make the LM querying tractable: prompting for confounders between pairs is natural and well-suited to the pairwise structure of PAG edges. We acknowledge that in general, a latent confounder may affect multiple observed variables simultaneously, and have stated this explicitly in the revised manuscript (Page 4, Notation and setup). Extending to multi-variable confounding is a direction for future work and is also mentioned now (Page 13, Limitations and Future Work).
> * **Availability of domain experts.** We agree this could be stated more precisely. We have revised this in the manuscript (Page 1, Abstract).

---

> ### Author Response · Authors · 2026-03-25
> **Author Response (2/2)**
>
> * **Bias in batch data.** Let us clarify this point. Our use of the term *selection bias* reflects the notion that in any single batch of a sequence of batch data the sample may not have been a random draw and thus may not be representative of the population distribution. We use selection bias in the same sense as [1] and also has its roots in an earlier seminal work [2]. Selection bias takes many forms including due to distributional shift; ours can be labeled a sample selection bias, where there is no shift in the distribution. In practice, how does sample selection bias play out? Consider the arrival of web data, weekly, for an online business. A single week's batch may not be representative, but cumulatively over many weeks we can get a representative sample. We have added an explicit definition early in the manuscript (Page 1) clarifying that "data-induced bias" refers to sample selection bias due to non-random draw from the population for any single batch. We assume there is no distributional shift.
> [1] Peter Spirtes, Christopher Meek, and Thomas Richardson. Causal inference in the presence of latent variables and selection bias. In Proceedings of the Eleventh conference on Uncertainty in artificial intelligence, 1995.
> [2] Heckman, James J. "Sample selection bias as a specification error." Econometrica: Journal of the econometric society (1979): 153-161.
> * **Uninformative priors.** We agree the analogy is imprecise and have removed the reference to uninformative priors now in Section 6.3.
> * **Hyperparameter selection.** We performed grid search over hyperparameters for all methods (NLPSCM and baselines), ensuring fair comparison. Regarding evaluation on a separate validation set: in our sequential batch setting, each batch is already small, and further splitting would degrade the quality of conditional independence tests. Instead, we report mean and standard deviation over 5 independent runs, providing a measure of robustness.
> * **Broader impact.** We thank the reviewer for this suggestion. We have added a broader impact statement to the revised manuscript (Page 13, Broader Impact) addressing the downstream use of learned causal models, including considerations around model misspecification and the need for careful validation before using discovered causal structures for decision-making.

---

> > ### Comment · Reviewer_2Teq · 2026-04-16
> >
> > Thank you for thoroughly acknowledging these requested changes and for the thoughtful replies. I am glad for the agreement on several of these points, and appreciate that these comments have been taken into account.
> >
> > All but one of the responses have fully addressed my concerns. Though not too critical, I still have some minor issues with **Bias in batch data**. First, let me comment that I am very familiar with the term *selection bias* as used in the sense of the cited works. I agree that selection bias does not represent a shift in the *underlying* population distribution, but it *does* represent a shift in the distribution from which the samples are drawn. For instance, in the web-data example, the underlying distribution of transactions may be $p(x)$, but taking samples from the first week of the year, one is drawing from the distribution $p(x | w = 1)$. The explicit definition will help, but I want to make sure that the prose reflects the distinction between shifts in the underlying distribution and shifts in the "observed distribution" due to selection bias.

---

> > > ### Author Response · Authors · 2026-04-16
> > >
> > > Thank you for the follow-up comment. We will add the following toward the beginning of Section 3 to provide more clarity, in addition to what we have already added in the 2nd paragraph of Introduction (marked in blue) in the current version.
> > >
> > > "We assume the underlying population distribution $p(X)$ is stationary across batches. However, we allow for selection bias, where each batch may not be drawn randomly from the population distribution. That is, for a batch $i$, $p(X \mid batch = i) \neq p(X)$."

---

### Review · Reviewer_rWjb · 2026-03-18

**Summary Of Contributions:**

The paper studies causal discovery when observational data arrive in sequential batches and reliable human experts are unavailable. It proposes NLPCSM, a hybrid framework that combines standard causal discovery with language-model-derived priors, while treating the LM as a noisy source of information rather than an oracle. A key idea is to use PAGs instead of DAGs so that ambiguity, partial orientation, and latent confounding can be represented explicitly. The method also includes budgeted LM querying and an extension to SEM parameter estimation, and the paper reports improved performance over several baselines on benchmark datasets.

Strengths

- The paper tackles a practically relevant setting that is less explored in prior LM-assisted causal discovery work. Much of the prior work discussed here assumes access to the full dataset or uses LM prompting in a more static setting, whereas this paper explicitly studies sequential batched data with noisy LM priors, which is a meaningful extension of the problem setting.

- The move from DAGs to PAGs is well motivated.

- Empirical results are generally strong, with useful ablations and reported mean ± std over runs.

Weakness

- The Bayesian framing seems overstated relative to the implementation. The paper writes down a posterior-style formulation, but the actual method is closer to a hybrid pipeline: run FCI, query the LM, aggregate answers in histograms, and apply entropy-based thresholds. That can still be useful, but it is not a fully specified Bayesian structure-learning procedure in the usual sense.
- Sequentiality is mostly simulated by splitting static datasets rather than evaluated in a truly sequential real-world setting. As a result, the experiments support performance under incremental access to static data more than they establish robustness in a genuinely sequential deployment scenario.

**Audience:**

Yes

**Audience Explanation:**

I think at least a subset of the TMLR audience would be interested in this paper’s findings. The paper sits at the intersection of causal discovery, hybrid data+LM methods, and reasoning under uncertainty, all of which are active topics in the TMLR community. In particular, researchers interested in using language models as imperfect scientific assistants rather than as oracles may find the paper’s framing useful.

The paper is also likely to be relevant to readers working on causal structure learning, especially because it studies how to combine standard causal discovery tools with noisy external prior knowledge and argues for using PAGs instead of DAGs in this setting. Even for readers who are not primarily focused on causal discovery, the broader question of how to integrate unreliable LM knowledge into principled learning pipelines is timely and potentially of wider interest.

**Claims And Evidence:**

No

**Claims Explanation:**

The paper provides reasonably convincing evidence for its main empirical claim that the proposed framework can improve causal structure recovery over the included baselines: the experiments cover multiple datasets, report mean ± standard deviation over 5 runs, and include useful ablations on thresholding and query allocation. In that sense, the practical usefulness of the method is supported.

However, some broader claims are not fully matched by the evidence. The paper motivates the method around selection bias / data-induced bias in sequential batches, but the formal setup assumes batches are sampled from the same underlying distribution, and the experiments mostly simulate sequentiality by splitting static datasets. So the evidence supports incremental batched causal discovery more than true robustness to biased sequential data.

**Requested Changes:**

- Clarify and narrow the central claim around sequential bias handling. The paper currently motivates the problem using selection bias / data-induced bias in sequential batches, but the formal setup assumes batches are sampled from the same underlying distribution and the experiments mainly simulate sequentiality by splitting static datasets. The authors should either add experiments and methodology that genuinely address biased sequential data, or narrow the claim to incremental batched causal discovery with noisy LM priors.
- The paper’s motivation is about practical causal discovery under sequential data arrival and limited expert availability, but the empirical section relies mostly on standard benchmark graphs and simulated batching obtained by splitting static datasets. The two User Level datasets help somewhat, but even there the evaluation is weakened by proxy ground truth inferred from another causal discovery method. To support the practical claims of the paper, I wonder if the authors could add at least one stronger real-world experiment with either known/validated causal structure, expert-backed edges, interventional checks, or a genuinely sequential deployment-style setting. Without that, the paper feels more like a promising benchmark study than a convincing demonstration of real-world utility.

---

> ### Author Response · Authors · 2026-03-25
> **Author Response (1/2)**
>
> We thank the reviewer for their insightful comments on the manuscript. We are encouraged that the reviewer found the sequential setting more relevant and a meaningful extension to the existing setup, the motivation for PAGs well justified, the empirical results to be strong, and the problem framing of interest to the TMLR audience. A recurring theme across the review is that some framing claims extend beyond the formal setup and experiments. We address the weaknesses and concerns holistically below.
>
> **Bayesian framing.** The point is well taken. In stating (Page 2, paragraph 2) that *"the causal structure discovery itself is Bayesian in that the beliefs about causal structure from data are updated iteratively by information drawn from an LM, as data arrive in batches,"* we were imprecise. We should have said that this belief updation across sequential batches is Bayesian inspired. We agree that the causal-structure learning algorithm of the framework does not maintain a full posterior distribution over causal structures and thus is not Bayesian in the classical structure learning sense. The intended use of the Bayesian-inspired framing (Eq. 3) is that it serves as a conceptual foundation for integrating noisy LM information over batches. In particular, we treat LM outputs as noisy, prior information and accumulate evidence across batches. We maintain edge-level belief summaries (via histograms) which can be interpreted as sequential estimation of edge-type distributions under noisy observations. These updates are analogous to Bayesian evidence accumulation and provide a principled, uncertainty-aware approximation at the edge level, even though they do not maintain a posterior over the graph space. NLPSCM produces a PAG augmented with uncertainty-aware edge beliefs that is updated efficiently across batches. We also note that our histogram-based formulation is closely related to recent work on semantic entropy in language models (Section 4.2) which similarly aggregates uncertainty across samples.
>
> We have revised the manuscript accordingly, and now describe the structure-learning algorithm as Bayesian-inspired (Page 2, Page 6, and Section 4.2) and make explicit that uncertainty is maintained at the edge level rather than as a posterior over graphs (Page 6 and Section 4.2). We have also mentioned the fully probabilistic inference over graph structures as an important direction for future work (Limitation and Future Work on page 13).

---

> ### Author Response · Authors · 2026-03-25
> **Author Response (2/2)**
>
> **Sequential setup, claims, and requested changes.** We note your concerns that some broader claims are not fully matched by the evidence. We address the concerns regarding sequentiality, simulated batching, support for broader claims, and the requested changes together, as they stem from a common observation.
>
> The goal of this manuscript is sequential causal discovery under batch data, rather than under distributional shift. The manuscript explicitly assumes stationarity (Sec. 3: "where each batch is a sample from the same underlying 'true' distribution $\mathcal{D}_i \sim \mathbb{D}$"), and all methodology and experiments are consistent with this.
>
> By “data-induced bias” we refer to selection bias (stated in Problem Statement of Section 3), rather than distributional shift across batches. Our use of the term *selection bias* reflects the notion that in any single batch of a sequence of batch data the sample may not have been a random draw and thus may not be representative of the population distribution. We use selection bias in the same sense as [1] and also has its roots in an earlier seminal work [2]. Selection bias takes many forms including due to distributional shift; ours can be labeled a sample selection bias, where there is no shift in the distribution. In practice, how does sample selection bias play out? Consider the arrival of web data, weekly, for an online business. A single week's batch may not be representative, but cumulatively over many weeks we can get a representative sample. In the revision, we have added an explicit definition early in the manuscript clarifying that "data-induced bias" refers to sample selection bias due to non-random draw of any single batch from the distribution (Page 1). We assume there is no distributional shift.
> [1] Peter Spirtes, Christopher Meek, and Thomas Richardson. Causal inference in the presence of latent variables and selection bias. In Proceedings of the Eleventh conference on Uncertainty in artificial intelligence, 1995.
> [2] Heckman, James J. "Sample selection bias as a specification error." Econometrica: Journal of the econometric society (1979): 153-161.
>
> In experiments, we use a controlled setup where a static dataset defines the ground truth and batches provide incremental access. This is a standard protocol for evaluating incremental access under a stationary data-generating process. Our use of metrics such as SHD and SID which require a fixed reference graph matches this protocol. Within this setting, the problem remains nontrivial: each batch is insufficient for reliable discovery, and the challenge is to accumulate structural evidence and allocate LM queries effectively across batches. The experiments with *only-data* methods showcase this problem, and ablations (Fig. 4) show that sequential optimization and dynamic thresholding yield consistent improvements over non-sequential baselines.

---

### Decision · Action_Editor_srMk · 2026-04-29

**Recommendation:** Accept with minor revision

**Additional Comments:**

Please check the following to make sure things are addressed properly:

- Confirm Figure 4 (right) legend colours are corrected in the final PDF; I do not see that to have been addressed
- Make sure the stationarity/selection bias clarification sentence appears at the opening of Section 3
- Be consistent with respect to "Bayesian-inspired" (rather than "Bayesian"), making sure it is applied consistently throughout the main paper and supplementary material if relevant, not only in the revised sections
- Check for typos, e.g. on page 18 in the supplementary penultimate sentence "the ground truth causal graph is shown in [Figure 2]spiegelhalter1993bayesian."
- Lastly please make sure the template is respected, highlights are removed, and any other "promises" have been incorporated into the final version, which will be checked and confirmed by the AC upon submission

**Audience:**

Yes

**Audience Explanation:**

The paper is well-suited to the TMLR audience. It focuses on causal discovery under batch data arrival and noisy LM priors that has been getting more attention in the ML literature recently. I believe it complements other work published at TMLR, and it is well within scope.

**Claims And Evidence:**

Yes

**Claims Explanation:**

This paper addresses causal discovery under two practical constraints, complementing existing work, i.e. data arriving in sequential batches subject to sample selection bias, and reliance on language models (LMs) as noisy surrogates for domain expertise. The proposed framework, NLPSCM, makes four core contributions: (i) a principled shift from DAG to PAG representations to natively accommodate structural uncertainty and latent confounding; (ii) a Bayesian-inspired sequential update scheme that accumulates LM-derived edge evidence via empirical histograms across batches; (iii) a bandit-style query selection policy that allocates a limited LM budget to the most informative edges; and (iv) an EM-based parameter estimation procedure that incorporates noisy LM priors on latent confounders. The framework is evaluated across six datasets spanning 5 to 37 variables, four LM families, and five metrics, consistently outperforming both data-only and data+LM baselines.

Some original claims about "Bayesian structure learning" and related topics that were not supported by the method and experiments presented were flagged by the reviewer(s) and corrected at the revision stage. Therefore, as the reviewers have suggested, this paper makes useful contributions, and the claims as presented in the revised version are supported by the method and results.